



# Assimilation of DAWN Doppler Wind Lidar Data During the 2017 Convective Processes Experiment (CPEX): Impact on the Precipitation and Flow Structure

Svetla Hristova-Veleva[1], Sara Q. Zhang[2,3], F. Joseph Turk[1], Ziad S. Haddad[1], Randy C. Sawaya[4]

[1]Jet Propulsion Laboratory, California Institute of Technology, Pasadena CA 91107 USA
[2]National Aeronautics and Space Administration, Goddard Space Flight Center, Greenbelt MD 20771 USA
[3]Science Applications International Corporation, McLean, VA 22101 USA
[4]University of California-Irvine, Irvine CA 92697 USA

*Correspondence to*: F. Joseph (Joe) Turk (jturk@jpl.caltech.edu)

**Abstract.** An improved representation of the 3-D air motion and precipitation structure through forecast models and assimilation of observations is vital for improvements in weather forecasting capabilities. However, there is little independent data to properly validate a model forecast of precipitation structure when the underlying dynamics are evolving on short

convective times scales. Using data from the JPL Ku/Ka-band Airborne Precipitation Radar (APR-2) and the 2-um Doppler Aerosol Wind (DAWN) lidar collected during the 2017 Convective Processes Experiment (CPEX), the NASA Unified Weather Research and Forecasting (WRF) Ensemble Data Assimilation System (EDAS) modeling system was used to quantify the impact of the high resolution, sparsely-sampled DAWN measurements on the analyzed variables and on the forecast when the DAWN winds were assimilated. Overall, the assimilation of the DAWN wind profiles had a discernible impact to the wind

field and the evolution and timing of the 3-D precipitation structure. Analysis of individual variables revealed that the assimilation of the DAWN winds resulted in important and coherent modifications of the environment. It led to increase of the near surface convergence, temperature and water vapor, creating more favorable conditions for the development of convection exactly where it was observed (but not present in the control run). Comparison to APR-2 and observations by the Global Precipitation Measurement (GPM) satellite shows a much-improved forecast after the assimilation of the DAWN winds –

development of precipitation where there was none, more organized precipitation where there was some, and a much more intense and organized cold pool, similar to the analysis of the dropsonde data. Onset of the vertical evolution of the precipitation showed similar radar-derived cloud top heights, but delayed in time. While this investigation was limited to a single CPEX flight date, the investigation design is appropriate for further investigation of the impact of airborne Doppler wind lidar observations upon short-term convective precipitation forecasts.



# 1    Introduction.

Atmospheric convection plays a major role in both weather and climate. However, the initiation of convection and the mechanisms through which it organizes and grows upscale, from isolated convective cells to organized mesoscale convective systems, still remain largely unknown (*Houze*, 2018). As a result, their representation in Numerical Weather Prediction (NWP) models remains inaccurate (*Peters et al.*, 2019; *Prein et al.*, 2015). At the same time, both of these processes, convective initiation and upscale growth, have very significant consequences for all scales of motion - from the smallest scales of turbulence to individual convective cells, mesoscale convective systems (MCS), tropical cyclones, planetary waves and climate (*Schumacher and Rasmussen*, 2020). The complex multidirectional transfer of energy, momentum, and water between scales and across the atmosphere has enormous impact on the generation of severe weather.

Convection is driven by complex multi-scale interactions. The key large-scale ingredients include the thermodynamic and dynamic states of the atmosphere, describing the presence of potential instability and the presence/absence of large-scale forcing needed to trigger upward motion and, eventually, to release the potential instability, leading to the initiation of convection. Large- and meso-scale convergence and the presence of atmospheric boundaries (in moisture and heat) can play the role of these triggers of convection. Occasionally, these individual storms organize and grow up-scale forming MCSs. But what factors lead to the MCS development on the first place? What are the dynamic and thermodynamic mechanisms by which individual isolated convective storms interact with each other (*Raymond et al.,* 2015), organize and grow upscale?

The cold pool dynamics are an important mechanism thought to facilitate the development of MCSs in the tropical atmosphere (*Chen et al.*, 2015; *Zuidema et al.*, 2017). These atmospheric boundaries can have significant impact on the initiation of new convection, affecting its intensity, organization and longevity. While pre-existing boundaries are ubiquitous in the atmosphere, once convection starts it generates its own convergence lines and boundaries as the convective overturning results in the creation of surface cold pools – regions that are colder than the surrounding air. These precipitation-generated cold pools create favorable conditions for forcing new convection along their leading edges where the warmer environmental air is being displaced by the horizontally-spreading colder air. As the initial convection progresses, these cold pools interact with each other and grow in size and intensity, leading to further system growth given the right environmental conditions - a positive feedback mechanism of self-organization and upscale growth.

The structure of the cold pools is controlled, in turn, by several factors. Two of them include the thermodynamic state of the environment (vertical distribution of temperature and humidity) and the vertical wind shear, affecting the turbulent mixing and entrainment near the storm edges. Horizontal transport and mixing of nearby dry air can weaken convection, by decreasing the buoyancy (*Schiro et al.*, 2020). At the same time, entrainment of dry mid-tropospheric air increases cloud evaporation, resulting in the development of stronger downdrafts and the build-up of the surface cold pools. A third factor that strongly impacts the


structure and the evolution of the cold pools, and the precipitating systems in general, is the microphysical characteristics of the precipitation (*Hristova-Veleva et al.*, 2020a), which strongly affects the evaporation rates. *Morrison et al.* (2012), among

others, found that numerical simulations with higher evaporation had stronger cold pools, faster propagation, larger storm size, greater updraft mass flux (but weaker convective updrafts at mid- and upper levels), and greater total condensation that compensates for the increased evaporation to give more surface precipitation. In turn, the structure and the intensity of the divergent near-surface cold pools modify the morphology of the convective systems. These joint processes affect the vertical growth and glaciation of water-abundant clouds, and further aggregation and organization of individual cumulus clouds into

much larger mesoscale convective systems (*Rowe et al.*, 2012; *Houze*, 2018).

While the overall processes responsible for these interactions have been identified for some time, their precise nature and interactions remains under-constrained by observations; in particular the uncertainty regarding convection and cloud processes directly results in much of the uncertainty in both weather and climate prediction. Further constraining the uncertainty in

convective cloud processes *linking 3-D air motion and cloud structure* through models and observations is vital for improvements in weather forecasting and understanding limits on atmospheric predictability. To date, there is little independent validation data to properly validate a model forecast of precipitation structure when the underlying dynamics are evolving on convective times scales.

Many years of NASA-sponsored airborne field campaigns have focused on the microphysical processes linking clouds, convection and precipitation, as well as ground validation, following the deployment of the Tropical Rainfall Measuring Mission (TRMM) satellite in 1997 and the Global Precipitation Measurement (GPM) mission (2014-current). These airborne campaigns featured narrow swath precipitation profiling radars, such as the JPL Ku/Ka-band Airborne Precipitation Radar (APR-2) (*Durden et al.*, 2012). However, the Doppler capability of these radars is intended for estimating the vertical Doppler

velocity within precipitating clouds, and are not capable of capturing vertically resolved observations of 3-dimensional wind structure in close proximity (10-km or less) to cloudy regions. A space-based Doppler wind lidar (DWL) capability has been envisioned as one means to overcome this observational shortcoming (*Okamoto et al.*, 2018; *Baker et al.*, 2014). The current Atmospheric Dynamics Mission (ADM)-Aeolus wind lidar (*Stoffelen et al.*, 2005) has been successfully collecting satellite-based line-of-sight profiles (*Lux et al.*, 2020), at a synoptic scale suitable for global numerical weather prediction (NWP) data

assimilation, rather than the spatial scale of cloud-resolving mesoscale models (*Šavli et al.*, 2018; *Horányi et al.*, 2015).

Previous DWL-based airborne campaigns lacked scanning Doppler precipitation radar capabilities on the same aircraft, whose data collection was synchronized with the DWL operations. During the May-June 2017 Convective Processes Experiment (CPEX), joint observations were collected from the APR-2 and the 2-um Doppler Aerosol Wind (DAWN) lidar (*Greco et al.*,

2020; *Kavaya et al.*, 2014) during approximately 100 flight hours of the NASA DC-8 aircraft (*Turk et al.*, 2020). The APR-2 radar operates at the same frequencies as the GPM Dual-Frequency Precipitation Radar (DPR). In particular, measurements



from the DAWN lidar provided high-resolution vertical profiles of the air motion in the environment close to where the clouds develop (*Zhang et al.*, 2018). To date, there has been relatively little analysis of the assimilation impact of airborne Doppler wind lidar data upon the joint evolution of the mesoscale model-forecasted 3-D precipitation structure together with the

associated 3-D wind field (*Cui et al.*, 2019).

In this manuscript, the impact of assimilating the high resolution, sparsely-sampled airborne DAWN measurements upon the forecasted precipitation structure are examined with the NASA Unified Weather Research and Forecast (NU-WRF) Ensemble Data Assimilation System (EDAS) modeling system (*Zhang et al.*, 2017; *Zhang et al.* 2013). A previous study of the impact

of assimilating DAWN data from CPEX was carried out by *Cui et al.* (2019), who examined how different assimilation methods affected the forecasted wind and 2-D precipitation structure inferred from the gridded GPM IMERG (*Tan et al.*, 2019) precipitation dataset. A unique aspect of this study is that both the horizontal and vertical evolution of the forecasted precipitation field is compared with near-simultaneous data from APR-2 radar data, and from DPR data from overpasses of GPM. This manuscript is a direct follow-on to the recently published manuscript by the authors (*Turk et al.*, 2020), which

describes in detail the APR-2 and DAWN data for the 10 June 2017 flight date investigated here. In particular, the graphics and discussion in the *Turk et al.* (2020) manuscript specifically tailored the DC-8 flight segments on June 10 into four one-hour defined segments. Each of those one-hour segments corresponds to the same assimilation time window used in the NU-WRF data assimilation cycles. The forecast impact is examined with and without (i.e., a control run) the assimilation of the DAWN wind profiles into the model. The role of the data assimilation process is to adjust the model forecast based on any

observed data, accounting for errors in the forecast and the observations. The assimilation impact is assessed in two steps. First, the forecasted precipitation field is compared between the NU-WRF control run and the DAWN assimilation run for each of the four one-hour segments. For both runs, the forecasted precipitation field is compared to the observed APR-2 precipitation structure. Times and areas where the assimilation demonstrated an improved 3-D representation of the precipitation structure are identified. In the second step, the model environmental state fields are compared between the control

run and the analysis, to determine how the model state (wind, temperature, moisture) changed in the model as a result of the assimilation of DAWN wind profiles. While this investigation and its conclusions are limited to a single CPEX flight date, the investigation design is appropriate for further investigation of the impact of airborne Doppler wind lidar observations upon short-term convective precipitation forecasts.

For the sake of not replicating a large number of figures in this manuscript, the discussion in this manuscript will make frequent reference to specific figure numbers from *Turk et al.* (2020) (full open access, so all can refer to it). To simplify the nomenclature, the term T2020 is used to cite that manuscript.



## 2        DAWN and APR-2 data during CPEX.

During CPEX, NASA DC-8-based airborne observations were collected from the JPL Ku/Ka-band Airborne Precipitation
Radar (APR-2) and the 2-µm Doppler Aerosol Wind (DAWN) lidar during approximately 100 flight hours.  The performance
of DAWN during CPEX is presented by *Greco et al.* (2020), and the complementary observations of APR-2 and DAWN
during CPEX, tailored to this 10 June 2017 case, are presented in T2020.  Therefore, only a brief description is provided here.

For CPEX, the APR-2 provided vertical air motion and structure of the cloud systems in nearby precipitating regions where
DAWN is unable to sense.  Conversely, DAWN sampled vertical wind profiles in aerosol-rich, no-cloud regions surrounding
the convection, but is unable to sense the wind field structure within cloud.  Figure 1 of T2020 shows the scanning operations
of both instruments onboard the DC-8 for CPEX.      APR-2 acquires simultaneous measurements of multiple parameters at
both Ku- and Ka-band (14 and 35 GHz), including co- and cross-polarized backscatter, and line of sight (LOS) Doppler
velocities of hydrometeors.  APR-2 scans cross-track to resolve the 3-D nature of precipitating clouds.  (For more recent field
campaigns, the APR-2 was modified into APR-3 with the inclusion of a W-band (94 GHz) radar, but this capability was not
available for CPEX).  APR-2 range (vertical) resolution of 37-m and cross-beam (horizontal) resolution of $\approx$ 800m at 9-km
distance are more than adequate to capture cloud features down to the resolution typical of high-resolution models, and
appropriate for comparison in the vicinity of DAWN wind profiles.

DAWN is NASA's highly capable airborne wind-profiling lidar with a 2-micron laser that pulses at 10 Hz.  DAWN can provide
high resolution (4-12 km in the horizontal and 35-150 m in the vertical) wind measurements in clear as well as partly cloudy
conditions.  The lidar scans in a conical pattern at a constant 30º elevation angle and collects line-of-sight (LOS) wind profiles
at up to five azimuth angles located at -45º, -22.5º, 0º, 22.5º and 45º relative to the aircraft flight direction.  Since these LOS
wind profiles view the local wind field from multiple azimuth angles, these LOS profiles are further processed to estimate the
profile of the horizontal wind components ($u, v$) at different pressure levels (*Greco et al.,* 2020).  In this presentation, these
profile data are used for the data assimilation impact studies.

## 3        NU-WRF configuration and simulations for the June 10 case.

The NASA-Unified Weather Research and Forecasting (NU-WRF) modeling system was used for all cloud-resolving
modeling and data assimilation tasks (*Zhang et al.*, 2017).  NU-WRF is an observation-driven regional earth system modeling
and assimilation system, including physics modules, a satellite data simulation unit (G-SDSU) capable of simulating modern-
era NASA satellite observations, including the GPM DPR Ku/Ka-band (14/35 GHz) equivalent radar reflectivity profiles, and
the GPM microwave imager (GMI) (*Matsui et al.*, 2014), and an ensemble data assimilation system that can assimilate
conventional state variables such as wind, temperature and moisture as well as cloud/precipitation affected microwave
radiances.  For this investigation, NU-WRF was adapted for assimilation of DAWN profile winds, and simulation of passive





MW brightness temperature (TB) at the 13 GMI channels (10.7 through 183.31 GHz), and the DPR equivalent radar reflectivity factor profiles at same frequencies as DPR and APR-2 (14/35 GHz).

In Figures 4-18 of T2020, the APR-2 flight tracks (overlaid upon nearby-time GOES-16 visible imagery), and associated APR-2 reflectivity and DAWN profiles are shown for each of the four one-hour assimilation windows, centered at 1900, 2000, 2100

and 2200 UTC.  Figure 4 of T2020 shows an example of the DAWN winds subsetted at one altitude (8-km) for the 10 June 2017 flight date, during the 1830-1930 UTC time period, which was just before the DC-8 entered the main area of interest. For simulations of this flight date, the NU-WRF EDAS model and analysis was configured as specified in **Table 1**.  The domain and flight area are shown along in **Figure 1**.

| | | |
|---|---|---|
| **Model** | Resolution | 9-km (domain 1, or d01), and 3-km (domain 2, or d02), with vertical 55 levels |
| | Time Steps | 45s (d01) and 15s (d02) |
| | Physics | Thompson 6 class microphysics, Grell 3D ensemble cumulus scheme |
| | Lateral boundary condition and forcing | NCEP GDAS |
| | Experiment period | 20170610 00Z - 20170611 00Z, including spin-up |
| **Analysis** | Algorithm | Ensemble maximum likelihood filter |
| | Control variables | wind, temperature, specific humidity, surface pressure, clouds and precipitation (liquid and frozen phase) |
| | Assimilation window | 1 hour |
| | Ensemble size | 48 |
| | Background error covariance | Flow-dependent, estimated from ensemble forecasts |
| | Observation types assimilated | NCEP conventional observations, DAWNv3 wind profiles |


**Table 1.  Configuration of the NU-WRF EDAS for use in this investigation.**

A 3-km inner grid was used for the comparisons with the APR-2 data.  In particular, the model assimilation cycle was hourly, incorporating all observations (conventional observations such as radiosondes) and DAWN winds within a ±30-min window,

centered on the top of the hour.  The one-hour forecast was then carried forward for the next hourly assimilation cycle. Precipitation is accumulated over one hour of model integration, and output at hourly intervals.  For example, a 0600 UTC assimilation cycle would incorporate all observations from 0530-0630 UTC.  The resultant one-hour forecast at 0700 UTC is used as the background in the next (0700 UTC) assimilation cycle.   The precipitation at 0700 UTC represents a one-hour integration from 0600-0700 UTC.

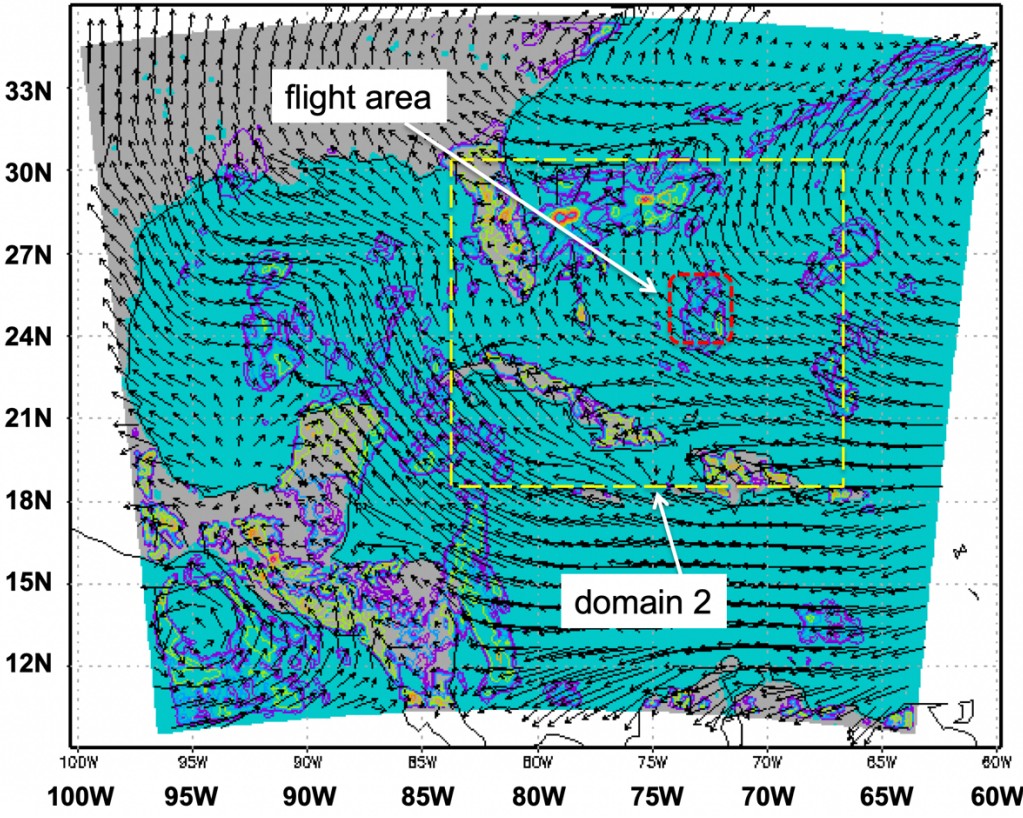


**Figure 1. Depiction of NU-WRF domain 2. 10-m winds are plotted from the control run at 1900 UTC. Domain 2 extends ~ 18N-30N, 84W-66W.**

The four panels of **Figure 2** show the cross-section of the DAWN wind zonal (*u*-component) vector wind field, processed for the one-hour assimilation cycles centered at each of the four one-hour assimilation cycles.    In general, the Doppler lidar-derived wind vectors are more abundant near upper levels (higher signal-to-noise ratio) and closer to the surface (more aerosols, larger backscatter), with a general loss of signal and less data in the mid-levels.    Areas of cloud contamination are shaded in blue color.  The NU-WRF EDAS was run in two modes: 1) a control run where only conventional observations (e.g.,

radiosondes, clear-sky radiances) are assimilated in the National Center for Environmental Prediction (NCEP) model that provides the initial boundary conditions; 2) a second data assimilation run where the DAWN wind profiles were assimilated in addition to the conventional observations.



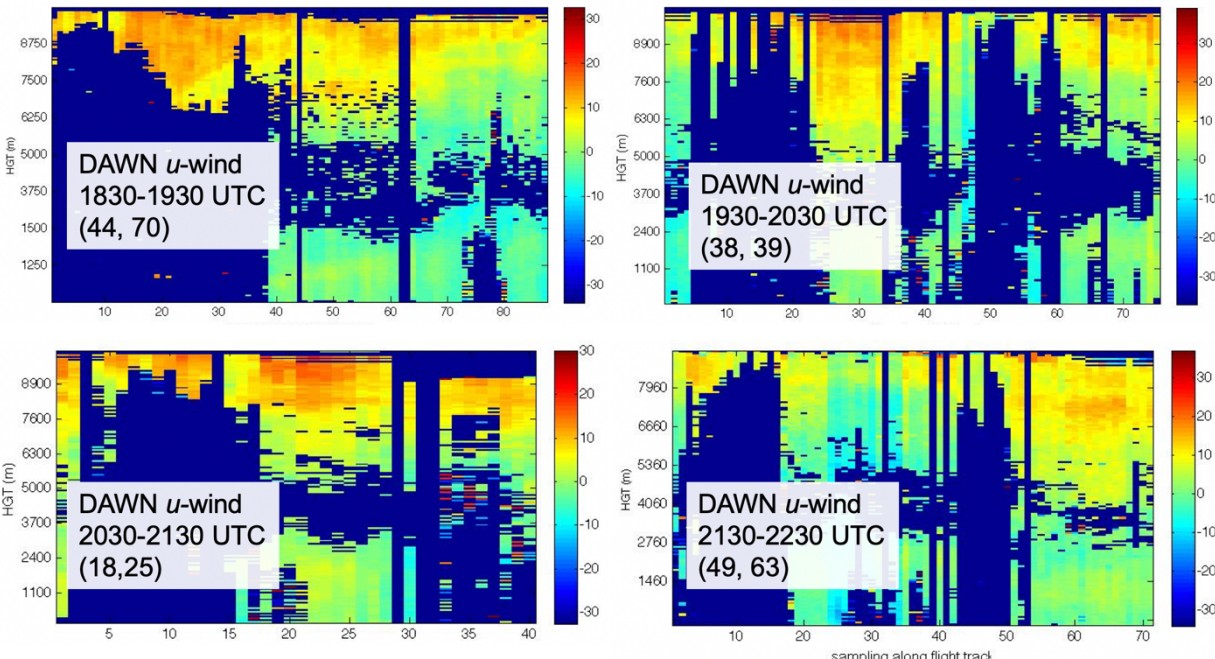

**Figure 2. Vertical cross section of the zonal (*u*) component of the DAWN wind profiles, preprocessed to (*u,v*), for the 10 June 2017 NU-WRF model impact study. In all panels, the x-axis represents one hour of DC-8 flight time, and y-axis extends from the ocean surface to ~9-km height. (Top row) 1900 and 2000 UTC data assimilation interval. (Bottom row) 2100 and 2200 UTC interval. The two numbers in parentheses indicated the number of DAWN wind vectors at 2 and 8-km height, respectively.**

**Figure 3** depicts the resulting model precipitation field forecast for the control run, showing the DC-8 flight track during each one-hour period. While some scattered precipitation develops in the periphery of the DC-8 flight patterns, no precipitation develops inside to the DC-8 flight track box patterns on this day. **Figure 4** shows the same set of figures, but after the DAWN DA. After the first data assimilation cycle (1830-1930 UTC), precipitation later develops inside the DC-8 box area between 1930-2030 UTC, and further intensifies into the 2030-2130 period and beyond. How well does this modeled precipitation compare to independent validation, in horizontal and vertical structure, and in timing?

**Control: Precipitation during 18-19 UTC**

**Control: Precipitation during 19-20 UTC**

**Control: Precipitation during 20-21 UTC**

**Control: Precipitation during 21-22 UTC**


**Figure 3.    Left to right from upper left: Precipitation from the control run, for the model forecast that was output at 1900, 2000, 2100, and 2200 UTC.  Each period represents a one-hour precipitation average (mm/h, contoured, according to the scale in Panel 1) and wind at 500m level (vector) are shown.  The domain is 21N-28N, 78W-70W.  The lines show the DC-8 flight tracks during each one-hour interval.**







**No Assimilation Yet: Precipitation during 18-19 UTC**  **After Assimilation: Precipitation during 19-20 UTC**

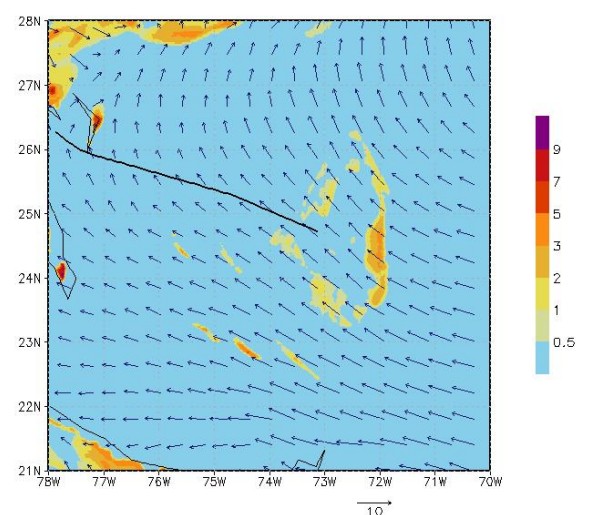

**After Assimilation: Precipitation during 20-21 UTC**  **After Assimilation: Precipitation during 21-22 UTC**

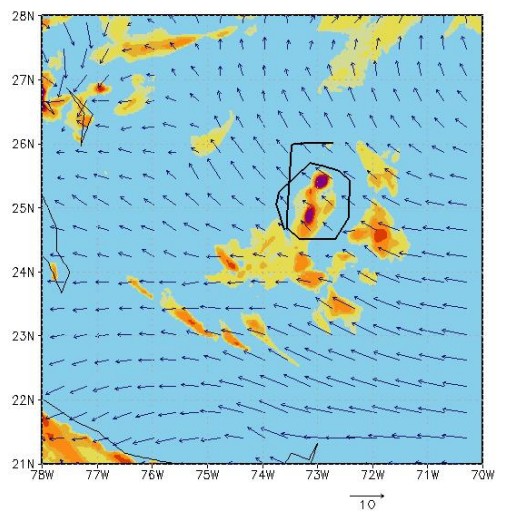
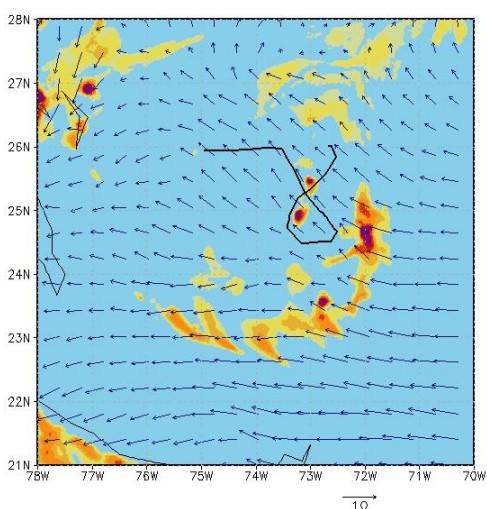

**Figure 4.  Same as Figure 3, but after the assimilation of DAWN wind vector profiles.**

Fortuitously, there was a GPM overpass directly over this region that occurred near 1852 UTC. **Figure 5a** shows the GMI

89H GHz image, showing the well-developed convection (TB < 200 K) to the north of the target area, but some indication

(only a few pixels in GMI) of developing convection inside of the target area (boxed area).   In order to provide resolution,

**Figure 5b** shows the DPR Combined Radar-Radiometer Algorithm (CORRA) (*Grecu et al.*, 2016) Ku-band only precipitation



rate inside of the boxed area (DPR has 4-km pixel size; individual pixels are plotted for detail). The developing convection occurs northeast of and near the end of the 1830-1930 DC-8 flight tracks (green dashed line). This location corresponds very

closely with the location of the modeled precipitation during 1930-2030 UTC (upper right panel of Figure 4). Therefore, location-wise, the modeled precipitation agrees well with independent observations observed by GPM GMI, even though it is present in the model ~1 hour later than in the observations, showing in the model following the assimilation of DAWN observations over the previous hour.

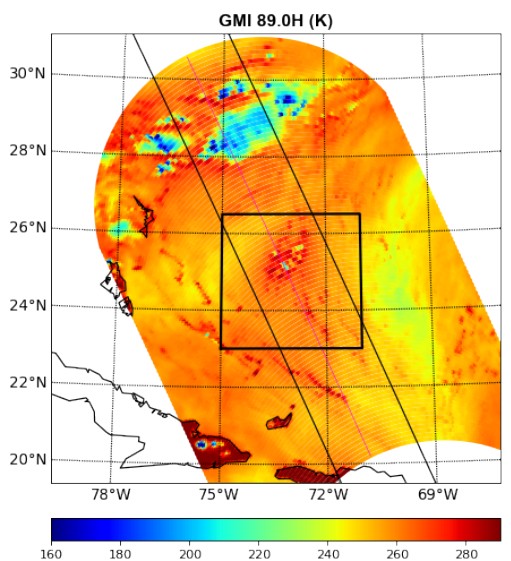
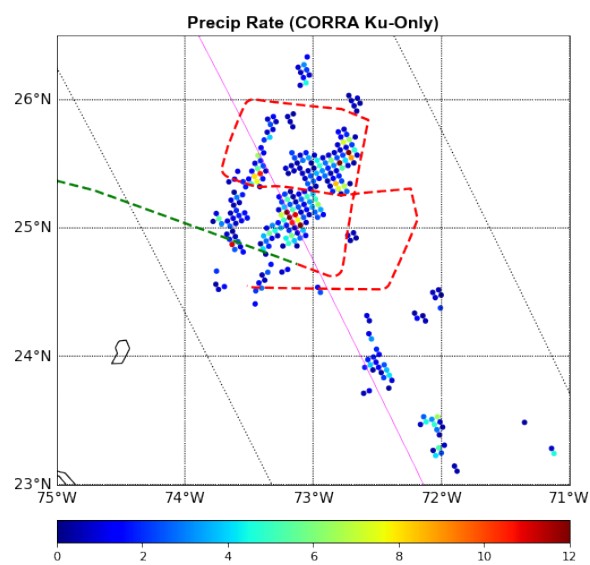


**Figure 5. GPM overpass near 1852 UTC on 10 June 2017. (Left). GMI 89H GHz channel (Kelvin units). (Right) Zoom-in to the precipitation rate estimated by the CORRA algorithm (mm hr⁻¹), over the box area shown in the left panel. In both panels, the black lines show the swath of the DPR Ku-band radar. On the left panel, the thin magenta line near the satellite sub-track denotes the DPR cross section shown in Figure 6. The green and red dashed lines indicate the DC-8 flight tracks between 1830-1930 and 1930-**
**2030 UTC, respectively.**

In Figure 5a, there is a thin magenta colored line that runs near the sub-track of the GPM satellite, that lies within the swath coverage of both the DPR Ku- and Ka-band radars. **Figure 6** shows the DPR cross section along this line. The resolution of

the DPR data has been averaged over a 3x3 area to match better with the resolution of the GMI 89 GHz channel (100 scan lines of GMI, corresponding to about 1000-km along-track distance). The top panel and middle panels of Figure 6 show the Ku- and Ka-band reflectivity profiles. The lower panel shows the trace of each of the 13 GMI channels under this same cross section. Near GMI scan 20, the radar tops are near 10-km, with significant attenuation of the Ka-band profile relative to Ku-band below the 4.5-km freezing level (blue dashed line). The developing cell in the boxed area of Figure 5a near (25N, 73W)

is near scan line 50, and the widespread convection above 28N is near scan line 20. The area near scan line 50 has less developed cloud above the freezing level, but also significant Ka-band attenuation relative to Ku-band. The passive MW TB



for channels < 89 GHz are not fairly similar for these two areas, but the ice scattering signatures at the GMI highest frequency channels (166 and 183.31 GHz) are more evident (significant TB depression) for the developed convection near scans 15-25.

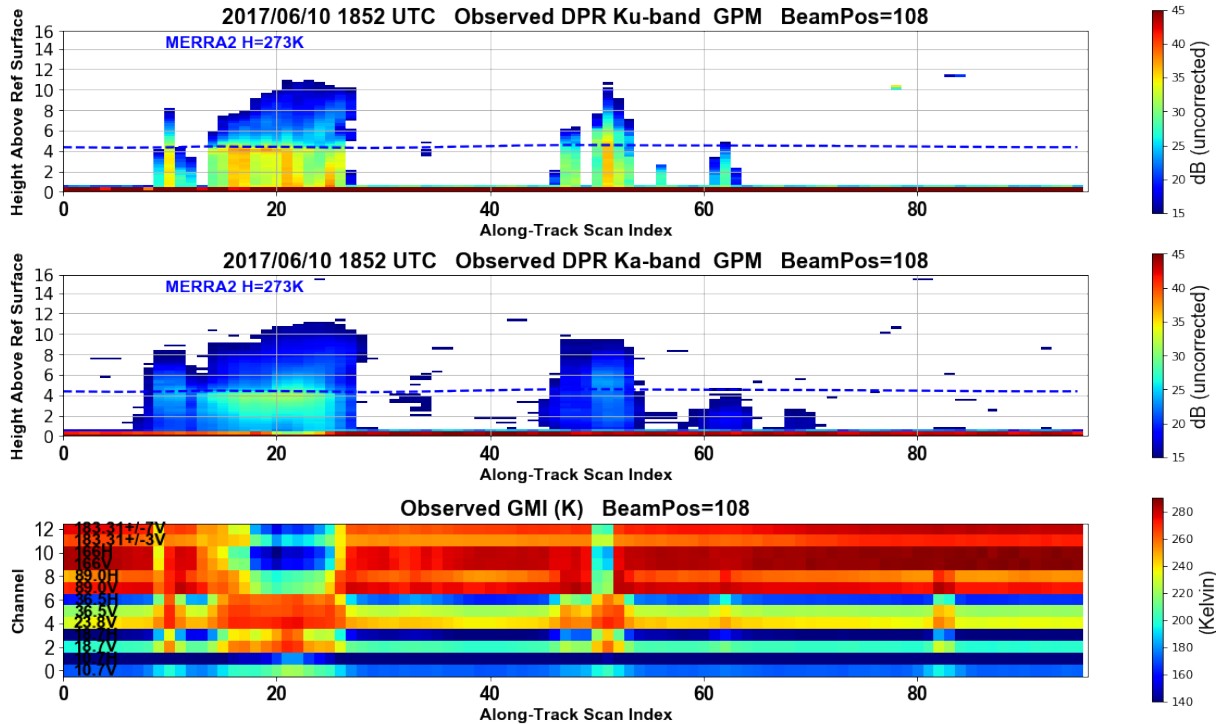


**Figure 6.  GPM overpass near 1852 UTC on 10 June 2017.  (Top). DPR Ku-band profile underneath the cross-section line indicated in the left panel of Figure 5.  Scan index=0 indicates northernmost location of the cross section.  (Middle) Same as top panel, but for the DPR Ka-band radar.  (Lower). Trace of each of the 13 GMI channels along this same cross section, ordered from lowest frequency (10.7 GHz) at the bottom to highest (183.31 GHz) at the top.**


These GPM radar and radiometer observations show good agreement with the *location* of the precipitation after the data assimilation of DAWN profile winds.  However, this alone is insufficient to explain what changed, in the model state variables, as a result of the assimilation.  Interpretation of the GPM data imply fairly high liquid water contents below the freezing level,

indicative of isolated, growing small-scale convection.  In the next section, the vertical structure of the NU-WRF simulations is analyzed and contrasted to profile characteristics from this DPR and actual APR-2 radar data.





## 4 Comparison of simulated radar profiles and model 2-D and 3-D fields.

As mentioned, NU-WRF provides post-processing options including an instrument simulator option, to forward-simulate satellite observations. This option was used to simulate radar observations at DPR frequencies and passive microwave TB at the 13 GMI channels. For purposes of this study, only the radar simulations are shown below.

### 4.1 Simulated DPR observations from NU-WRF.

The simulation of the DPR Ku-band radar observations using the microphysics, water vapor and temperature structure from the NU-WRF analysis at 2000 UTC is shown in **Figure 7**. The overall extent of the image is the same as the panels in Figures 3 and 4, and the boxed area corresponds to the geographical area shown in Figure 5b. The color scale refers to the maximum Ku-band reflectivity (dBZ) encountered in each model grid vertical column. The developing cloud near (25.5N, 72.9W) is well-coordinated in location with that shown in Figure 5b, and with peak reflectivity in excess of 30 dBZ. The strong convection along 28N is apparent, similar to what was observed by GMI. To further analyze the simulated DPR vertical structure, the black dashed lines indicate locations for N-S and E-W cross sections from two locations, one in the NE corner near (26.8N, 76.8W) and another in the boxed target area near (25.5N, 72.9W).

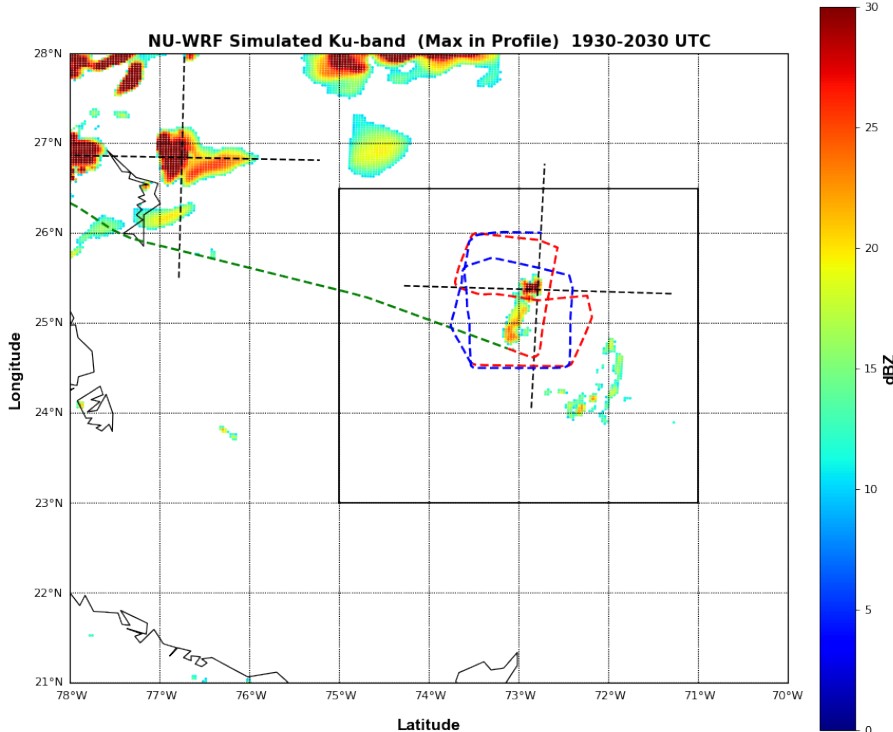

**Figure 7. Plan view of NU-WRF simulated Ku-band reflectivity (shown as maximum reflectivity in the profile) from the NU-WRF analysis at 2000 UTC. The dashed lines indicate the locations of the E-W and N-S cross sections shown in Figure 8, crossing at locations (26.8N, 76.8W) and (25.5N, 72.9W). The green, red and blue dashed lines indicate the DC-8 flight tracks between 1830-1930, 1930-2030 and 2030-2130 UTC, respectively.**






**Figure 8** shows, from top down, N-S and E-W cross sections for the simulated Ku-band reflectivity profiles, and then the Ka-band reflectivity profile simulations. Figure 8a (the left four panels) is associated with the convection near (26.8N, 76.8W) just inside of the NU-WRF model domain 2 shown in Figure 1. Simulated DPR reflectivity tops are near 10-12 km, also with

strong Ku- and Ka-band attenuation. The strong Ka-band attenuation is similar to what was noted from actual DPR observations during the earlier time (1852 UTC) GPM overpass (Figure 6).

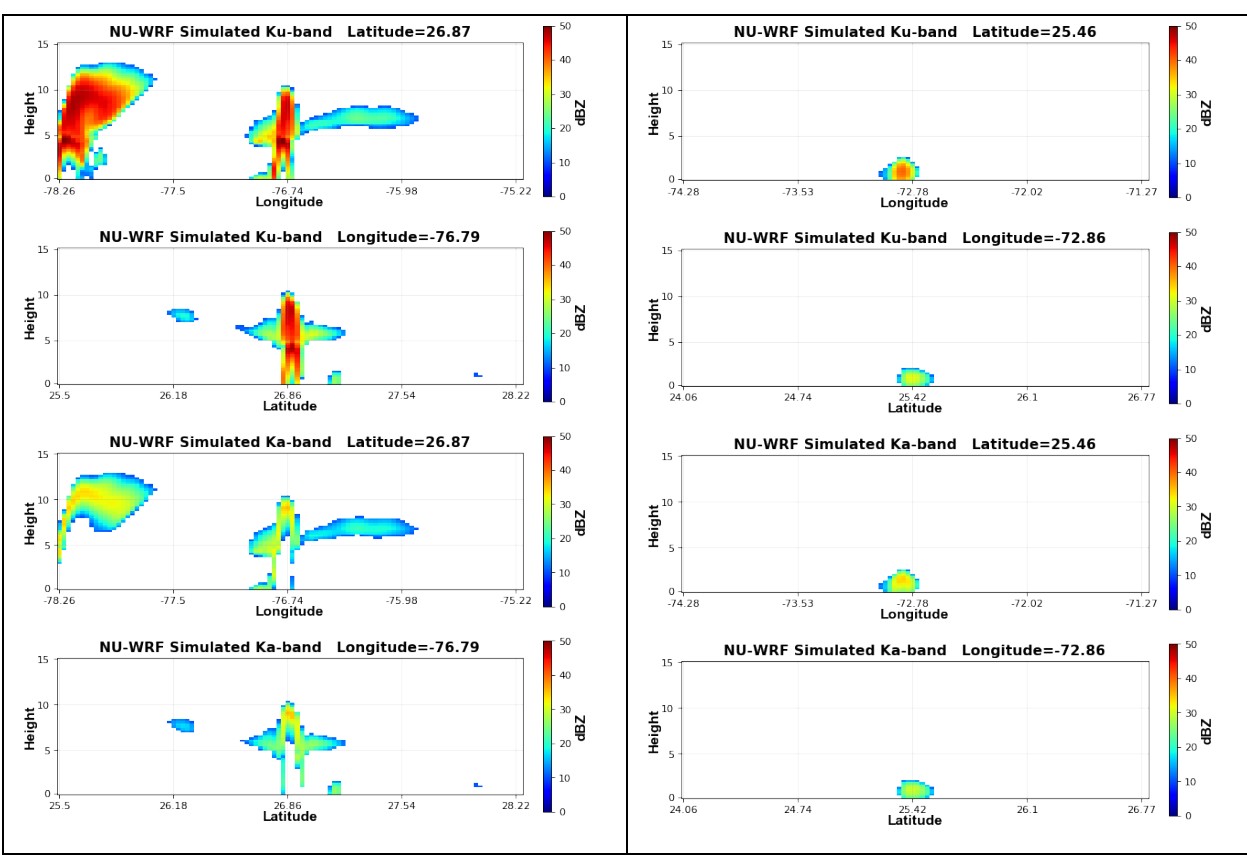

**Figure 8. Convection at the 2000 UTC analysis. Left column= NU-WRF simulated DPR profiles under the black dashed lines**
**crossing at (26.8N, 76.8W) in the box in Figure 7. From top to bottom in: Vertical profile of Ku-band reflectivity under the E-W line (Latitude=26.8N), Ku-band under the N-S line (Longitude=76.8W), and then these same two again but for the Ka-band simulation. Right column= same as left, but for the NU-WRF simulated DPR profiles under the black dashed lines crossing at (25.5N, 72.9W) in the box in Figure 7.**






Figure 8b (right four panels) is associated with the cross sections through the developing area in the boxed area of Figure 7 near (25.5N, 72.9W).   While the actual developing convection shows > 35 dBZ max reflectivity in the N-S Ku-band cross-section), NU-WRF modeled these as shallow clouds, limited to < 3-km vertical extent in simulated DPR cloud tops.  For the N-S cross-section, even within these very shallow all-liquid clouds, ~10 dB Ka-band path attenuation is present relative to Ku-

band, in accord with the associated DPR overpass indicating the presence of very high liquid water content.

To compare these simulated profiles with APR-2 observed profiles, **Figure 9** shows an APR-2 cross section (Ku- and Ka-band) between 2000-2010 UTC, about midway through the DC-8 flight segment (red dashed line) in Figure 7.   Essentially, Figure 9 is a close-up of the APR-2 profile shown in Figure 12 of T2020, but showing both radar frequencies.   The clouds in

this area are more mature and developed than what NU-WRF had forecasted, with Ku-band cloud top exceeding 8-km level (owing to the APR-2 radar configuration, the radar was unable to sense in the 1.8-km zone below the DC-8 flight altitude).  Strong differential attenuation (i.e., Ku- minus Ka-band difference increasing closer to the surface) was noted near scan 170, indicative of high liquid water content below 4-km.

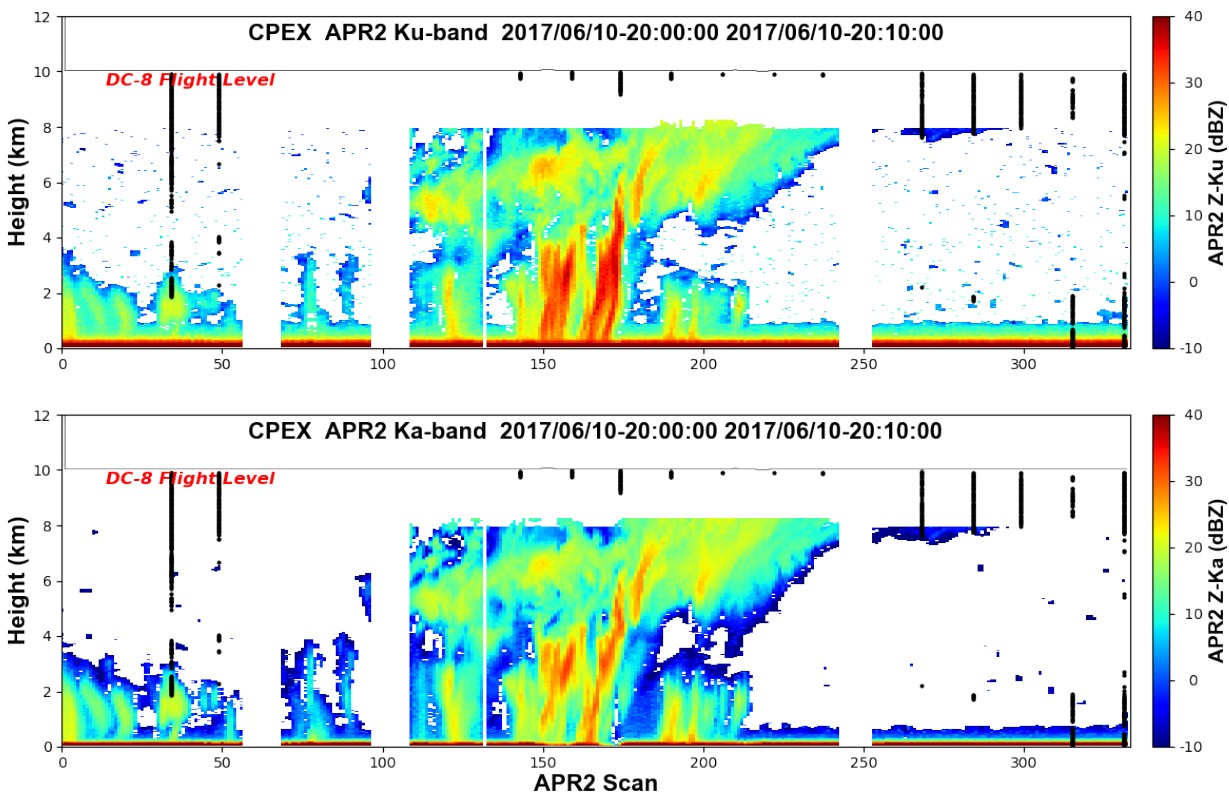


**Figure 9.  APR-2 measured radar reflectivity vertical profiles, for the 10-minute DC-8 flight period between 2000-2010 UTC.  (Top) Ku-band.  (Below) Ka-band.  The spacing between each APR-2 scan is 360-m.  The black dots indicate DAWN wind profile locations.**





**Figure 10 and Figure 11** are identical in layout to Figures 7 and 8, respectively, but for the for the NU-WRF 2100 UTC

analysis. The locations for the associated N-S and E-W cross sections are depicted in Figure 10 (black dashed lines), showing

that the precipitation during this time has evolved into two main active regions of precipitation inside of the boxed area, with

peak Ku-band reflectivity exceeding 30 dBZ. In Figure 11a, the associated vertical cross-sections show the rapid growth of

the cloud near (25.5N, 73W), with 45 dB radar tops near 10-km, more in accord with the APR-2 structure during this time (see

Figure 15 in T2020). In Figure 11b, the cloud near (25N, 73.3W) has intensified to near 45 dB but has developed to only 5-

km radar tops, in both the N-S and E-W directions. The Ka-band attenuation is severe in both cells, with the radar signal being

lost (below simulated DPR detection limits) before reaching the surface.

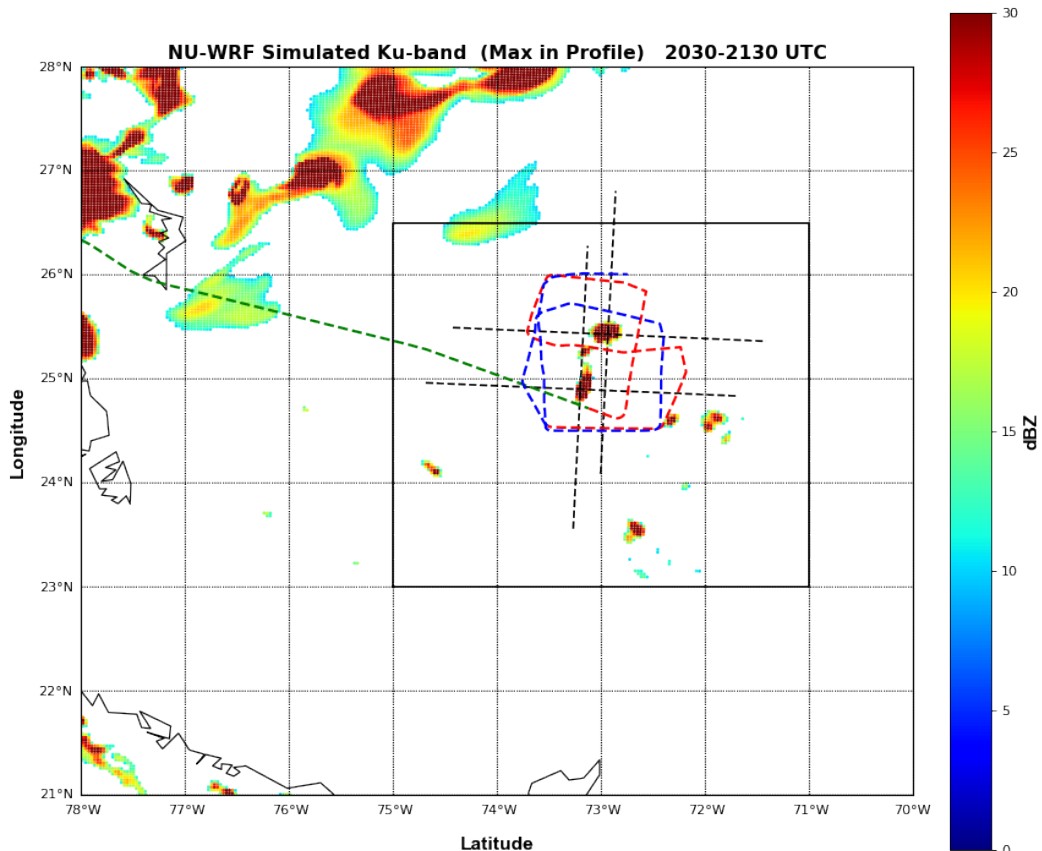

**Figure 10. Plan view of NU-WRF simulated DPR Ku-band (max reflectivity in the profile) at the 2100 UTC analysis. The dashed**
**lines indicate the locations of the E-W and N-S cross sections shown in Figure 11, crossing at locations (25.5N, 73W) and (25N, 73.3W). The green, red and blue dashed lines indicate the DC-8 flight tracks between 1830-1930, 1930-2030 and 2030-2130 UTC, respectively.**



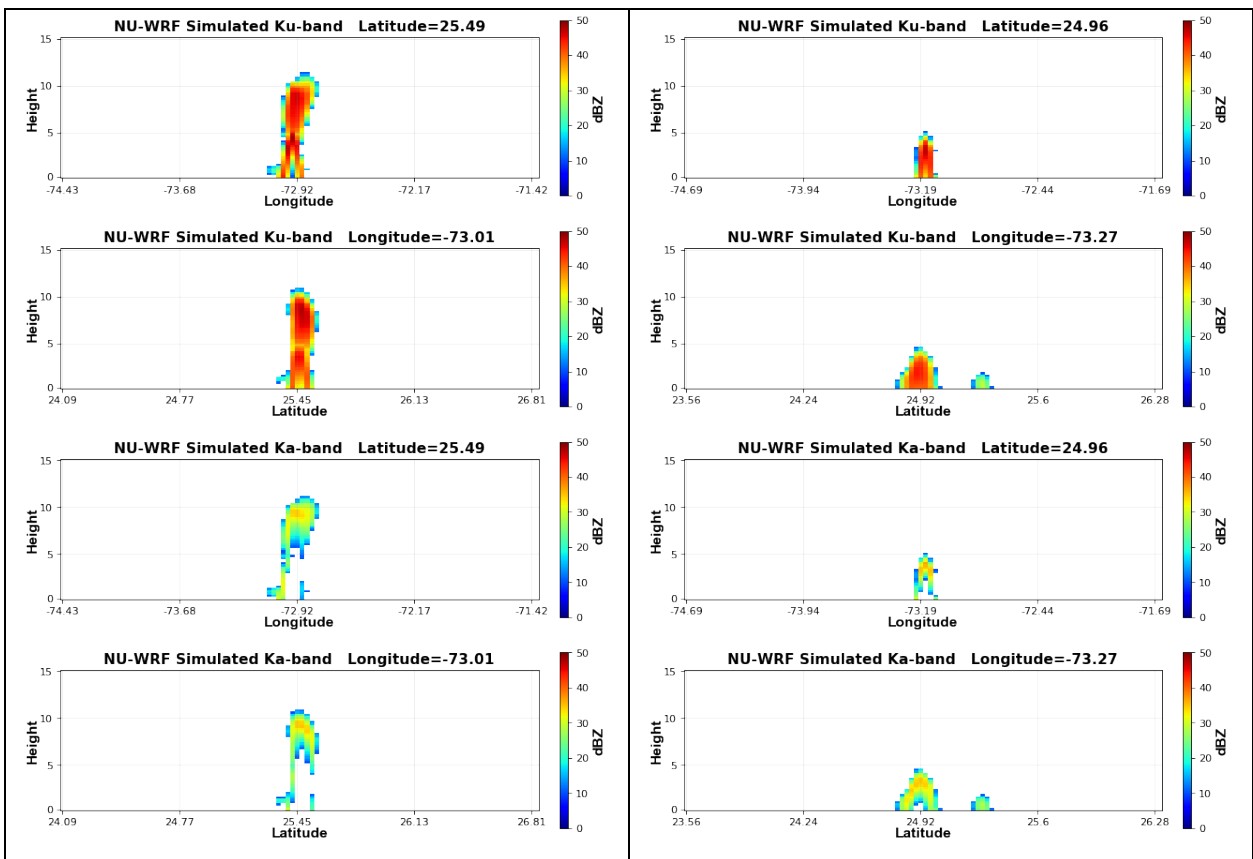

**Figure 11. Convection at the 2100 UTC analysis. Left column= NU-WRF simulated DPR profiles under the black dashed lines crossing at (25.5N, 73W) in the box in Figure 10. From top to bottom: Vertical profile of Ku-band reflectivity under the E-W line (Latitude=25.5N), Ku-band under the N-S line (Longitude=73W), and then these same two again but for the Ka-band simulation. Right column= same as left, but for the NU-WRF simulated DPR profiles under the black dashed lines crossing at (25N, 73.3W) in the box in Figure 10.**

To compare these simulated profiles with nearby observed profiles, **Figure 12** shows an APR-2 cross section (Ku- and Ka-band) between 2115-2125 UTC, near the end of the DC-8 flight segment (blue dashed line) in Figure 10 (essentially, Figure 12 is a close-up of the APR-2 profile shown in Figure 15 of T2020). The clouds in this area are more mature and developed than what NU-WRF had forecasted, with Ku-band cloud top exceeding 8-km level and especially strong differential attenuation (below 4-km level) near scan 200.

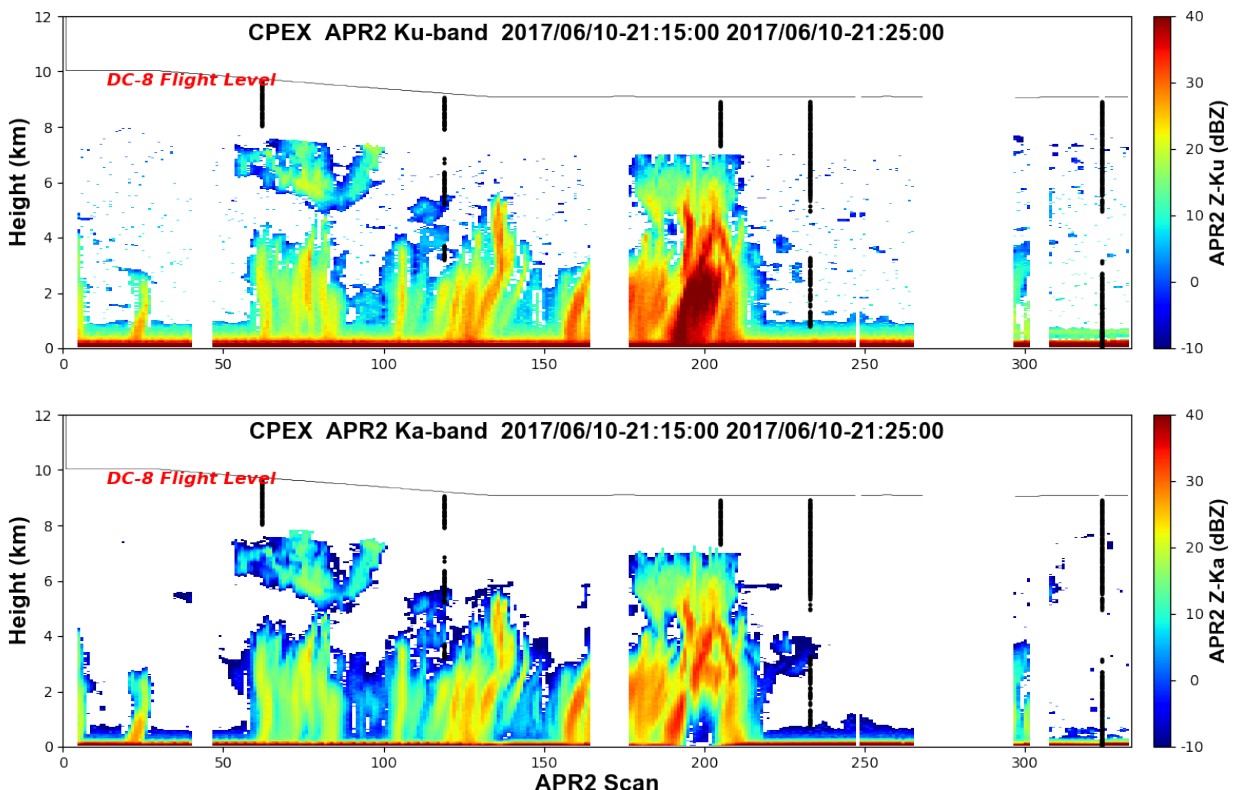

**Figure 12. APR-2 measured radar reflectivity vertical profiles, for the 10-minute DC-8 flight period between 2115-2125 UTC. (Top) Ku-band. (Below) Ka-band. The spacing between each APR-2 scan is 360-m. The black dots indicate DAWN wind profile locations.**


In summary, the control NU-WRF simulations failed to produce precipitation inside the box that was sampled during the June-10 flight mission. In contrast, according to both APR-2 and the GPM satellite observations, convection was observed in that box. After the assimilation of the DAWN winds, NU-WRF developed convection in the places where it was observed.

However, the development of the convection was delayed by about 1-hour. While the NU-WRF simulations well-represented the location of the developing precipitation (even though delayed in time), the associated growth in the heights of vertical precipitation structure also evolved slower. The NU-WRF simulated Ku/Ka-band radar tops did not reach the vertical development noted by APR-2, but they had better agreement to APR-2 cloud structure in the 2030-2130 period (an hour behind the observed precipitation). Hence, the assimilation of the DAWN winds resulted in the development of clouds and

precipitation, even though delayed, where it was observed. Interestingly, NU-WRF succeeded in producing the observed characteristics of the clouds and precipitation – predominantly shallow, non-glaciated clouds with high liquid water content, noticed in the strongly attenuated Ka-band radar profile.



### 4.2    Impact of the DAWN data assimilation on the model wind, temperature and moisture structure: Analyzing the analysis increments

The analysis above has focused upon the change to the convective structure that occurred following the assimilation of the DAWN wind profiles in the NU-WRF EDAS.   As noted in Figure 4, the assimilation of the DAWN winds, beginning with the 1900 UTC assimilation cycle, produced subsequent precipitation in the area where it was observed, whereas the control run produced no precipitation in the same region.  A relevant question is how the assimilation of the DAWN winds contributed to the subsequent development of precipitation in the area where it was observed in reality.  While this study is not of a scope

to fully answer these questions, one can compare the environmental state (structure of wind, potential temperature and water vapor) that was produced by the control model forecast, to the analysis produced after the assimilation cycle, to address the impact of the DAWN data assimilation on modifying the initial environment in which the subsequent convection develops. Indeed, the assimilation of the DAWN winds, even at a single time step, produced a very significant impact to the associated wind, temperature and moisture structure, as further illustrated below.


To examine, the direct impact of the DAWN data assimilation on the wind structure of the model is presented. The environmental wind shear conditions that were present at the time of the first assimilation period (a ±30 min assimilation window centered at 1900 UTC) are illustrated in Figure 7 in T2020, which depicts the vertical wind shear conditions inferred solely from DAWN profiles during this period.   There was sustained directional wind shear between 2-km and 8-km levels in

the area west of 73W, oriented from west to east, whereas between 2-km and 6-km the shear was weaker and oriented more south to north.  For a particular example, Figure 2 above (top left panel) illustrates the observed vertical profile of the zonal component of the wind during the 1900 UTC assimilation cycle, which incorporated observations between 1830-1930 UTC. These wind conditions were provided to the NU-WRF EDAS in the DAWN assimilation run, and absent in the control run. Figure 13 presents the analysis increments (assimilation-minus-control) introduced in the vertical profiles of the zonal ($u$) and

meridional ($v$) wind components after the 1900 UTC assimilation cycle.  A close look suggests the assimilation of the DAWN winds resulted in a decrease of the zonal wind shear immediately next to the convective development, at the end of the DC-8 track (80-90 km distance).  This is manifested by more positive valued increments near the surface and more negative increments at upper levels (above 6.25 km).  The meridional component increments suggested the reverse; i.e., an increase in meridional shear in the 80-90 km range (near the subsequent precipitation).  This is manifested by the more negative increments

at that range near the surface versus the more positive increments at the same range but at higher altitudes.

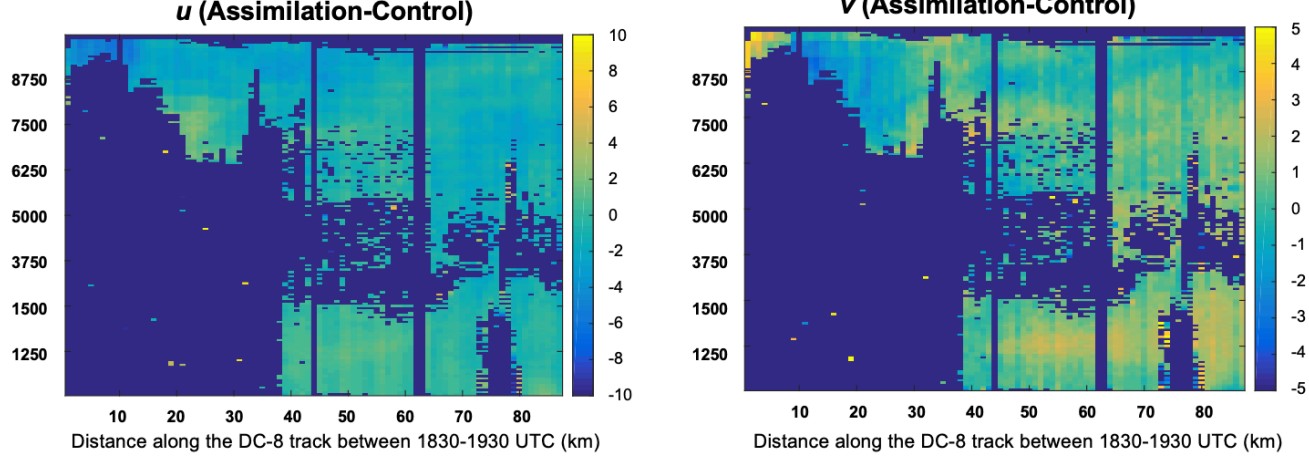

**Figure 13. Profile of the analysis increment (assimilation run at 1900 UTC -minus- control run at 1900 UTC) of the (left) zonal (*u*) and the (right) meridional (*v*) wind components of the flow along the DC-8 flight track as shown in Figure 3 (top-left panel). Units are in m s$^{-1}$.**


**Figure 14** shows the NU-WRF 500-m height model wind field after the first assimilation cycle (centered at 1900 UTC). The wind increments are broken down into their zonal and meridional components and plotted as a difference field (assimilation-minus-control). The red curves indicate the approximate boundary (high gradient) between the negative- and positive-valued

contours, with a focus on the area where the subsequent precipitation would develop in the data assimilation run. The zonal winds (left panel) reveal positive differences to the west of this boundary (stronger westerly winds) and negative differences to the east (stronger easterly winds), both indicative of stronger low-level convergence of the zonal wind in the assimilation run versus the control, i.e. – a stronger zonal forcing after the assimilation. The meridional winds (right panel) reveal positive differences (increments) to the south of the boundary line (stronger southerly winds) and negative differences to the north

(stronger northerly winds), also indicative of stronger low-level meridional convergence in the vicinity of the subsequent development of precipitation. Therefore, the assimilation of the DAWN winds modified the low-level wind field in such a way as to strengthen convergence (both zonal and meridional, almost at the same location) in a narrowly-focused zone. This line of convergence provided favorable dynamical conditions to promote further vertical cloud development.


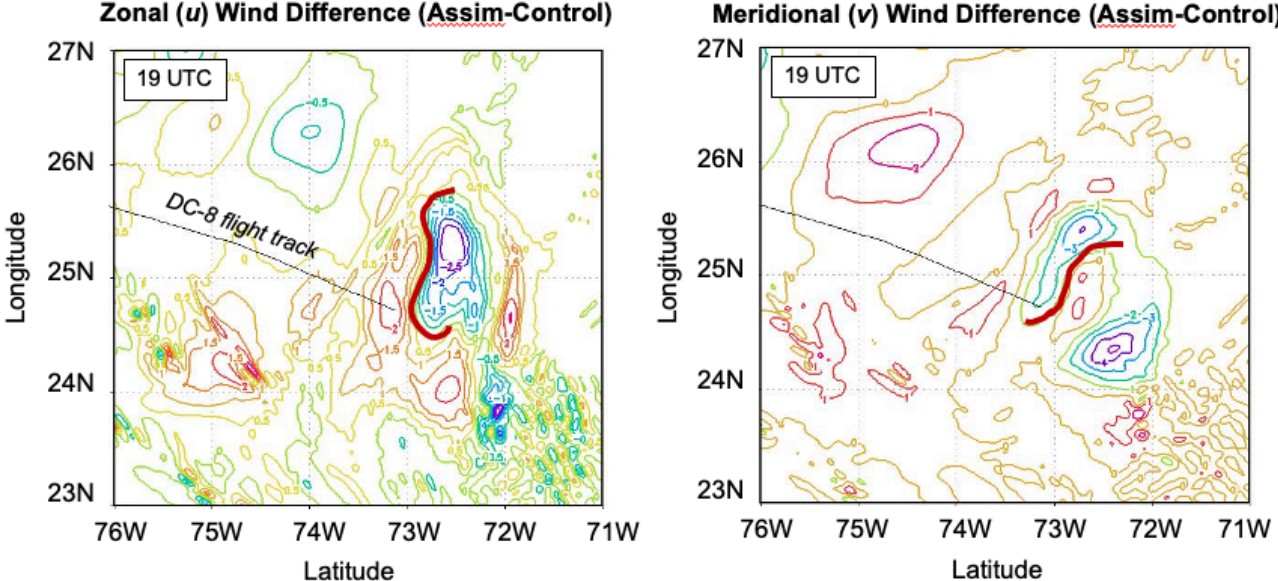

**Figure 14.** **500-m height wind field difference (assimilation minus control, in units of m s⁻¹) that resulted after the first data assimilation period centered at 1900 UTC.** **Left: Zonal (*u*) wind component difference, in m s⁻¹.** **Right: Meridional (*v*) wind component difference, in m s⁻¹.** **The red curves in each panel shows the approximate boundary between negative and positive contours, in the vicinity of where the precipitation eventually developed.** **The DC-8 flight segment during this time is shown.**

In a similar fashion to Figure 14, **Figure 15** shows the NU-WRF water vapor mixing ratio increments (left panel) and the

potential temperature increments (right panel) resulting from this same DAWN assimilation cycle (centered at 1900 UTC),

also plotted as a difference field (assimilation-minus-control, i.e., an analysis increment).    The region of strongest positive

increments (contours) is shown in the red shaded area.   It can be seen from Figure 15 that the assimilation of the DAWN

winds during this time produced positive moisture and temperature increments in highly-overlapping areas. While only new

wind data were assimilated (no new moisture data), the resulting increments in moisture and temperature were produced

through the background error covariances, generated by the model ensemble.   Both the higher moisture and the warmer

temperatures resulted in enhancing the convective potential in these regions.



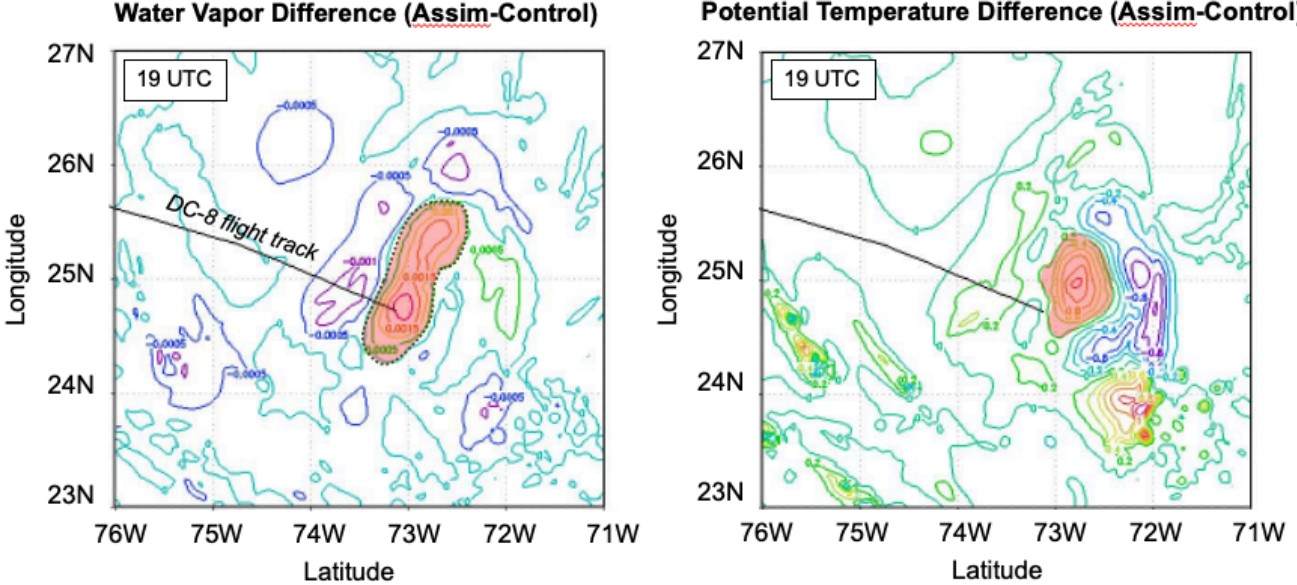

**Figure 15. Thermodynamic increments (assimilation minus control) that resulted after the first DAWN assimilation period centered**
**at 1900 UTC. Left: Water vapor mixing ratio difference, in units of kg kg⁻¹. Right: Potential temperature difference, in Kelvin.**
**The shaded red area denotes the area of the highest contour levels (strongest positive increments), in the vicinity of where the**
**subsequent precipitation developed in the assimilation run. The DC-8 flight segment during this time is shown.**

Figure 16 provides additional observational evidence of the enhanced moisture in the area of the observed convective

development. This evidence comes from combining a number of retrievals of the total precipitable water (TPW) and passive

MW rain index (RI) as provided by a variety of NASA, NOAA and EUMETSAT satellite systems. The rain index is a multi-

channel index combining brightness temperatures (TB) in the 10-90 GHz range (*Hristova-Veleva et al*., 2020b). These

quantities are illustrated in the JPL CPEX portal (https://cpexportal.jpl.nasa.gov) that combines satellite and airborne

observations with model forecasts, specifically tailored for CPEX. The portal options provide interactive visualization and

on-line analysis tools to help understand tropical convection processes (*Hristova-Veleva et al*., 2020b). Figure 16 depicts the

RI for the same GPM overpass as in Figure 5, with associated high satellite-derived TPW conditions in this identical area (not

shown).





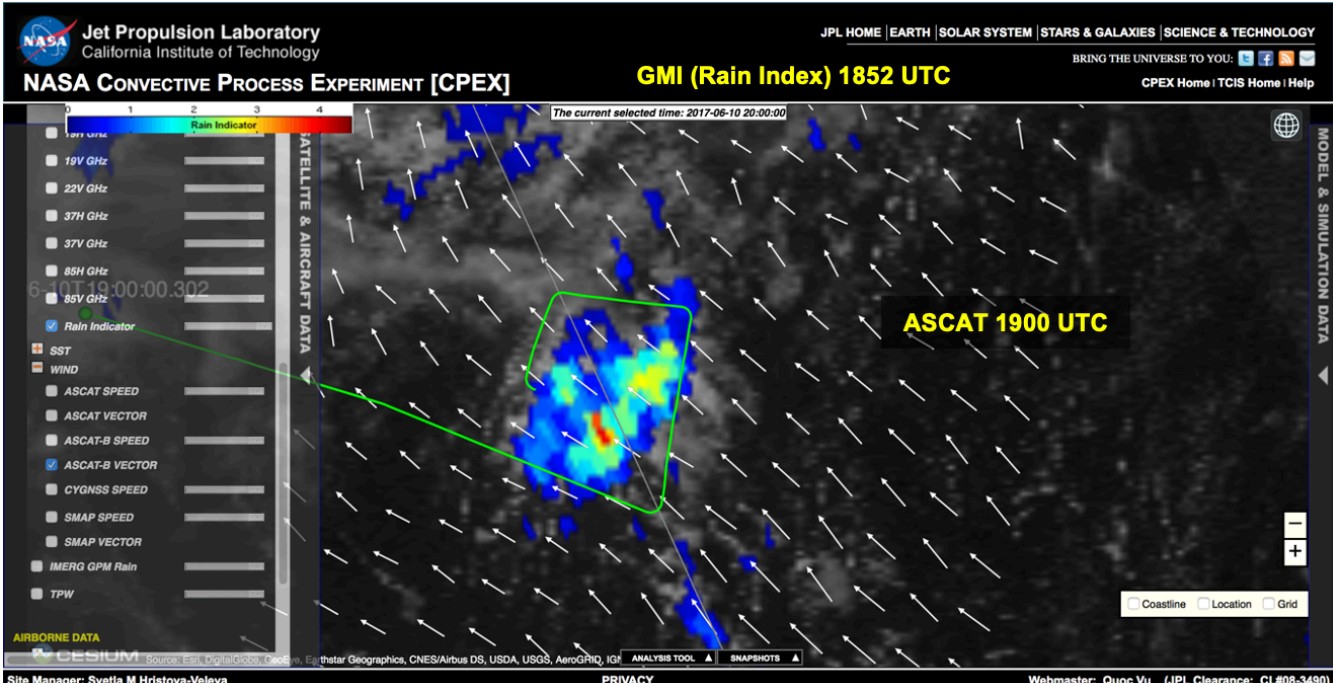

**Figure 16. Depiction of the rain index derived from GMI at 1852 UTC, shown in the JPL CPEX data portal overlaid on top the ASCAT-derived ocean surface winds at 1900 UTC.**

These overall conditions (enhanced near-surface wind convergence, accompanied by enhanced low level moisture and temperature) that resulted from the DAWN assimilation provided favorable conditions for eventual vertical development. After the assimilation, the subsequent forecast produced precipitation where there was none in the control run. Where the precipitation developed, it appeared more organized. **Figure 17** provides a conceptual interpretation of the underlying drivers and consequences. This figure summarizes differences in precipitation in relation to the analysis increments introduced by the assimilation of DAWN data during the 1900 UTC assimilation cycle. The features in Figure 14 (zonal and meridional wind convergence) and in Figure 15 (increased moisture, temperature) are co-registered in Figure 17a. The analysis increment produced surface convergence, co-located with increased moisture and temperature. Figure 17b overlays these features on top of the resultant precipitation in the model integration interval (1900-2000 UTC period), following this period of DAWN data assimilation, shown earlier in Figure 4b. These dynamic and thermodynamic components increased in a very coherent way, strongly suggesting that the resultant precipitation in the model integration interval was the consequence of the DAWN-assimilation inducing increasing of the convective potential exactly where the precipitation was observed. This resulted in convective initiation and the subsequent development of precipitation where there was none in the control run. From this, can one identify the important self-aggregating processes that allowed this initial convection grow upscale to produce the radar extensive area of precipitation, within the box of the flight area (DC-8 flight line in Figure 17b).



470

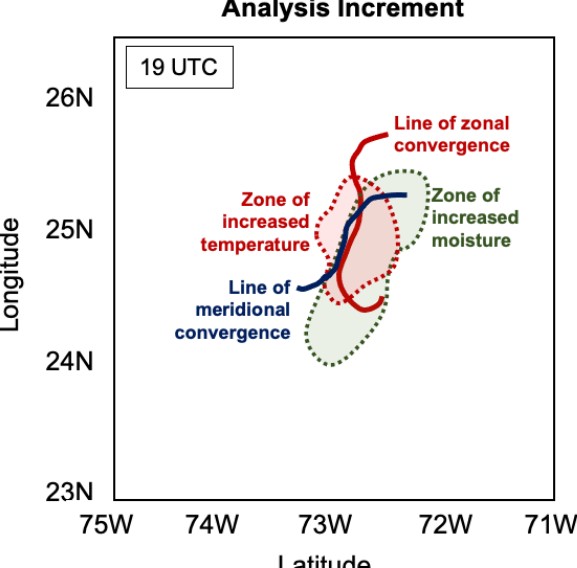
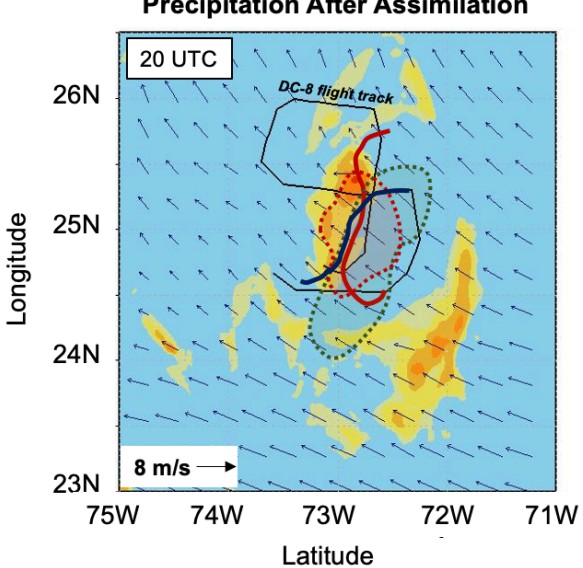

**Figure 17.** Relating the analysis increments produced by the assimilation of the DAWN winds to their impact on the precipitation field. Panels are a zoom-in (75W-71W, 23N-26.5N) of the area shown in Figures 3 and 4. (Left) Analysis increments (assimilation-control) as a result of the 1900 UTC assimilation cycle. Marked are the lines of enhanced convergence (zonal in blue; meridional in purple) shown in Figure 14, and the zone of increased moisture (green) and the zone of increased temperature (red) shown in Figure 15. (Right) Superposition of these same lines and contours onto the precipitation analysis at 2000 UTC (representing a one-hour model integration from 1900-2000 UTC) which was shown in Figure 4b. The DC-8 flight segment during this time is shown.

As precipitation develops it produces precipitation-loading-driven downdrafts that lead also to the entrainment of drier mid-level air - ready to evaporate the precipitation when the two come in contact. When precipitation evaporates, it further enhances the downdrafts, making them more negatively buoyant because of the loss the latent heat needed for the evaporation. When these downdrafts reach the surface, they spread out, being colder and denser than the surrounding air. This leads to the creation of the so-called cold pools – areas that are colder than the surrounding air, causing them to spread radially outward (*Schlemmer and Hohenegger*, 2014) as density currents. The cold pools created by the individual downdrafts interact with each other and the mesoscale flow organizes them into bigger entities. In turn, these precipitation-induced cold pools lead to the initiation of new convection along their leading edged by creating favorable conditions of forced lifting of the environmental air, affecting the location, strengths and organization of the convection that develops later on. As this environmental air is warmer and has more moisture, the induced lifting comes as an additional benefitting component, further improving the chances for the development of new convection and precipitation.

490





These mechanisms behind the formation and dissipation of the cold pool process (*Zuidema et al.*, 2017; *Grant and van den Heever*, 2016), and its identification in cloud resolving model simulations (*Drager and van den Heever*, 2017) are beyond the scope of this investigation. Here, the role of the cold pools in terms of their structure and relationship to the precipitation development is addressed. The control run and the assimilation run are compared and contrasted in terms of the near-surface temperature anomalies that develop. The two model forecasts are compared using the 2-m air temperature anomaly (difference from the initial state) as a footprint of the cold pool structure. **Figure 18** shows the 2000 UTC analysis 2-m temperature anomaly (shaded), precipitation (contoured in thin red lines) and wind at 500-m level (vectors). The thick solid red line denotes the approximate boundary of the cold pool that was detected in dropsonde observations taken during the June 10 flight (*Zipser and Rajagopal*, 2018).

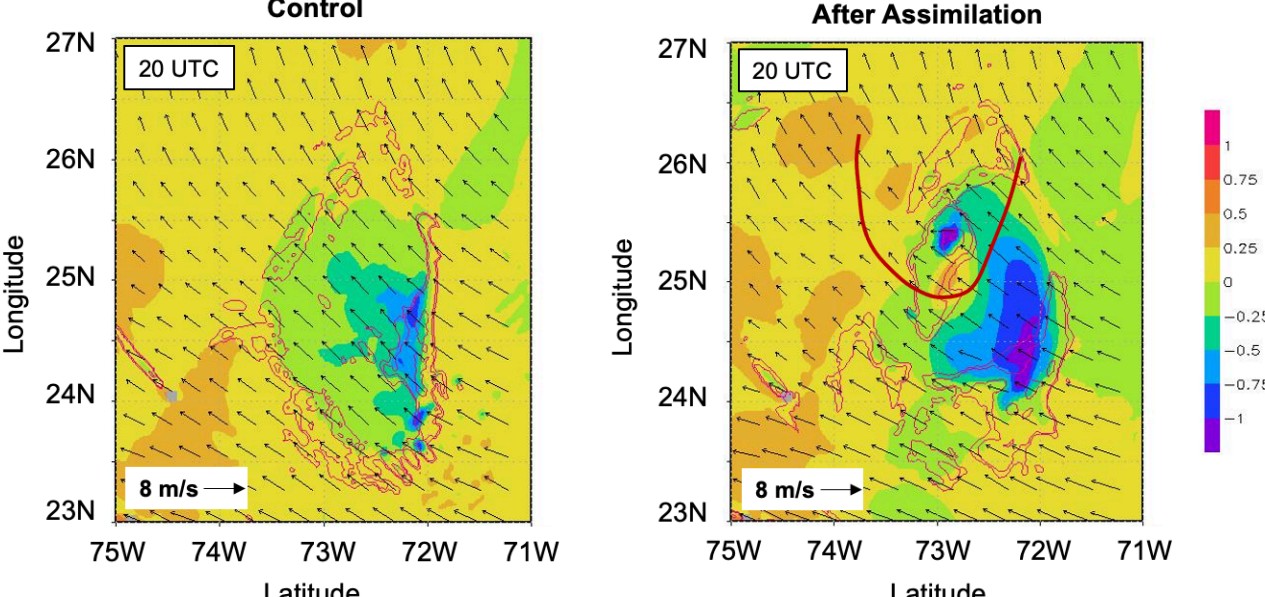

**Figure 18.** Cold pool structure and development showing the impact of assimilating DAWN winds at 1900 UTC. The cold pools are depicted by the 2-m temperature anomaly (from the initial state) at 2000 UTC, after 1 hour of model integration following the 1900 UTC assimilation. Overlaid are the cold pools, in Kelvin (shaded), precipitation during the 1900-2000 UTC model integration (contoured in thin red lines) and wind at 500-m level (vectors). (Left) Control run. (Right) After assimilation of the DAWN wind profile data. The thick red line indicates the approximate boundary of the observed cold pool created by downdrafts, as analyzed from dropsonde data by *Zipser and Rajagopal* (2018).

Both control and assimilation run show cold anomalies (cold pools) that are closely related to the precipitating areas, as it should be expected. However, the assimilation run shows much more intense and bigger cold pools. Two maxima are of interest. First, the smaller one to the north that is closely related to the precipitating area in the assimilation run (Figure 18b) that was not present in the control (Figure 18a). This cold pool, while not very big in areal extent, is closely related to the one



observed in the dropsonde data as marked by the thick red line. Second, the much bigger, better organized and stronger cold
pool found to the south-east. This extensive area of cold anomalies is related to the much bigger and organized precipitating
system there. Interestingly, this area of organized precipitation shows further signs of upscale growth as revealed by the
precipitation structure at the later 2000-2100 UTC and the 2100-2200 UTC periods revealed in Figure 4.

## 5        Conclusions.

This manuscript has presented the results of the impact to the forecasted precipitation structure that resulted when DAWN
wind vector profiles were assimilated by the NASA NU-WRF EDAS. This study is a direct follow-on to the recently published
manuscript by the authors (*Turk et al.*, 2020), which describes in detail the DAWN observations during each of the one-hour
periods used in the assimilation, and the APR-2 data for the 10 June 2017 flight date used for this impact study. The study
focused on (a) understanding whether (and if so, how) the assimilation of the DAWN winds impacted the subsequent
development of convection and precipitation, and (b) determining what environmental factors were modified by the
assimilation and understanding how would they have possibly impacted the development of precipitation.

The impact was examined from two directions. First, the structure and timing of the model precipitation field was examined
relative to that observed by the APR-2 radar data collected coincidently with the DAWN data. The Goddard SDSU instrument
simulator was used to simulate DPR (Ku/Ka-band) radar profiles for direct comparison to APR-2 and a GPM overpass that
occurred during the first data assimilation cycle. Second, the structure of the NU-WRF model winds, temperature and moisture
was contrasted between the model control run and the model data assimilation run. With these prognostic variables, the pattern
of convergence of moister air was examined, to explain the role of the thermodynamics in the evolution of the resultant model
precipitation horizontal and vertical structure, and how the vertical structure evolved in time relative to the APR-2 observations.

During 1830-1930 UTC time interval, dense DAWN observations were sampled in the surrounding environment, notably in
the cloud-free region just W-SW of the area of interest (Figure 4 in T2020), showing fairly strong wind shear between the
upper and lower levels. This is the time interval just preceding the onset of the precipitation within the DC-8 coverage area
noted in the model data assimilation run. While the NU-WRF simulations well-represented the location of the developing
precipitation in the subsequent 1900-2000 UTC period, the associated growth in the heights of vertical precipitation structure
evolved slower, with better agreement to APR-2 cloud structure in the 2030-2130 period. In accord with actual DPR data
collected earlier (1852 UTC), NU-WRF produced shallow, non-glaciated clouds with high liquid water content, noticed in the
strongly attenuated Ka-band radar profile.



Assimilation of the DAWN winds in NU-WRF EDAS, even at a single time step, produced a very significant impact. Analysis of the model wind field showed that as a result of the assimilation, the lower-level convergence was enhanced in this same general region. It resulted in modification of the near surface convergence (increased), the 2-m air temperature (increased) and the water vapor (increased). Analysis of individual variables revealed that the assimilation of the DAWN winds resulted in important and coherent modifications of the environment. It led to increase of the near surface convergence, temperature

and water vapor, creating more favorable conditions for the development of convection exactly where it was observed (but not present in the control run). The realism of the forecasted precipitation structure was shown by comparisons with nearby satellite and aircraft observations. Comparison to observations from APR-2 (and a fortuitous GPM satellite overpass) shows a much-improved precipitation forecast after the assimilation of the DAWN winds – development of precipitation where it was observed but not present in the control, and more organized structure where the precipitation eventually developed. Most

importantly, the assimilation produced a much more intense and organized cold pool, similar to one detected in a separate analysis of the dropsonde data collected during the DC-8 mission flight on that day. It is noted that a similar result was noted by *Cui et al.* (2019) in their DAWN assimilation study (modification of the near surface wind convergence field), taken from a different modeling system and two CPEX flight dates different than the date studied here.

While encouraging, these findings represent a single case. A longer assimilation period, and more flight dates, are needed to establish any repeatable impact from which to draw conclusions. This is challenging, given limited duration flight dates and fairly short (typically 3-4 hours) aircraft on-station times that encounter convection in its early formation stages. Future research can also address many other important questions. Are cold pools more effective at initiating new convection made in a variety of different environmental conditions, including variable aerosol loading? (aerosol effects were not addressed

here). In each case, one can relate the environmental parameters to the strength and the structure of the cold pools, and then their ability to generate and continuously support new convection at their leading lines, eventually resulting in an upscale growth of the system. The future NASA-ESA CPEX-AW field campaign, currently planned for 2021, will also provide the opportunity to fly APR-3, DAWN and the HALO lidar (*Bedka et al.*, 2020) synchronized with ADM-Aeolus orbits, in the eastern Atlantic where African easterly waves interact with the Saharan air layer (*Zipser et al.*, 2009). The investigation design

presented here, based on the availability of concurrent precipitation radar observations, is appropriate for further investigation of the impact of airborne Doppler wind lidar observations upon short-term convective precipitation forecasts.

**Data Availability**

The DAWN profile data (ASCII text format), APR-2 data (HDF5 format) and NU-WRF model output fields (binary format)

are available from the authors upon request.



**Team list**

**Author Contribution**

FJT carried out the data processing of DAWN, GPM and APR-2 data. SQZ performed the NU-WRF data assimilation and modeling simulations. SHV, ZSH and RS performed model post-processing and model diagnosis.

**Competing Interests**

The authors declare that they have no conflict of interest.

**Acknowledgements**

The work contained in this presentation was carried out at the Jet Propulsion Laboratory, California Institute of Technology, under a contract with NASA. © 2020 all rights reserved. Support from NASA under the Weather and Atmospheric Dynamics
program is recognized. The authors gratefully acknowledge the DC-8 flight support team, the DAWN data processing efforts by Steve Greco at Simpson Weather Associates, and the CPEX Co-Investigators Ed Zipser and Shuyi Chen.

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
