# Peer review of "Assimilation of DAWN Doppler Wind Lidar Data During the 2017 Convective Processes Experiment (CPEX): Impact on the Precipitation and Flow Structure"

_Atmospheric Measurement Techniques, 2020_

## Referee Comment (RC1) · Anonymous Referee #1 · 5 Jan 2021

Review of: Assimilation of DAWN Doppler Wind Lidar Data During the 2017 Convective Processes Experiment (CPEX): Impact on the Precipitation and Flow Structure

Authors: Svetla Hristova-Velev, Sara Q. Zhang, F. Joseph Turk, Ziad S Haddad, Randy C. Sawaya

Summary This paper demonstrates the impact of assimilating DAWN airborne Doppler wind lidar observations on a high resolution, convection allowing model simulation of a developing cluster of convective cells over the tropical Atlantic. Significant impacts

to the initial analysis and subsequent forecast are shown, where convection was not present in the control but was relatively robust (though delayed a bit in time) and in general agreement storm vertical structure depicted by APR-2 and GPM DPR observations. The paper is straightforward and well-written. I feel that it is acceptable for publication after the following generally minor comments are addressed.

Comments Line 45, "in the first place" instead of "on the first place"

Line 79, times -> time

Line 98, Zhang et al 2018 seems to be related to the NOAA P-3 DWL, not DAWN.

Line 101, recommend defining what you mean by "sparsely-sampled"

Line 135, I'm guessing that the environments weren't 100% cloud free given the tropical environment being sampled. DAWN has the ability to pulse through some tenuous cloud as well. So I'd recommend this being rephrased as "wind profiles in aerosol-rich, clear or broken cloud regions surrounding the convection". You address make a similar statement to what I recommend below in lines 146-147, so it would be good to be consistent with your statements.

Line 147, recommend "off-nadir" rather than "elevation" which could be interpreted to be 60 degrees coming out of the aircraft by some unfamiliar readers

Line 155-158, is GPM data at all assimilated into the boundary condition analyses or coarser grids in your simulation (i.e. is it part of the NCEP "conventional observations" in Table 1, and "clear sky radiances" in line 190)? Or are the simulations entirely independent of GPM, and putting DAWN into the mix made the MCS simulation look a lot more like GPM than the control.

Line 188, Blue seems like it is depicting both areas with little aerosol and cloud obscuration. You should consider depicting the two sources of DAWN data dropout with differing colors to better inform the readers.

[Figure]

**AMTD**

Figure 2, vertical axis should be made the same across all 4 panels to be consistent and to enable comparisons across segments

Page 13-14, I don't feel that you've adequately explained the nuances associated with Ka and Ku band reflectivity. Some places Ku only is shown (Fig 5), others both bands are shown. It would be helpful to provide a couple sentence explanation of the chaaracteristics of these multiple wavelength data.

Line 316, is this 1.8 km "blind spot" common for APR-2 and -3 instruments, or was this due to a instrument specific scan mode setting at the time of this flight?

Line 328, delete the 11a that has been crossed out

Figures 7-12, though it is demonstrated that the control generated almost no precip where there should have been some, I recommend you explicitly state somewhere that the profiles are from the run with DAWN assimilation.

Figure 16, I'm not seeing observational evidence of enhanced moisture, seems to look more like precipitation features rather than a moisture image. I recommend you reconsider this figure and the text that goes with it.

Figure 18, should the temperature anomaly in color shading in Figure 18b be aligned in some sense with the cold pool boundary? I am seeing a disconnect and am confused. I realize that the dropsondes revealed a cold pool there but I'm guessing a more substantial cold pool would have been found near the precip in the lower right quandrant of the domain had you released dropsondes there. Perhaps the red line is not particularly important because it confuses the interpretation of Fig 18b,

Line 547, the sentence beginning with "It resulted in modification of near surface convergence" seems a lot like line 549, "it led to increase of the near surface convergence". Was this an inadvertant repeat, or are there important nuances in the transition between the two sentences that I'm not understanding?

Line 550, you state that convection was not present in the control run, and then line

553 you state that precipitation was not present in the control run. Along the lines of the previous comment, perhaps you could review the paragraph beginning with line 545 and streamline the messaging. It's a very important paragraph to your summary so its necessary to be clear with your statements.

Line 560, you note that this was a single case, but the final sentence of the previous paragraph, you note that Cui et al found similar results as you, in that convection was better predicted through DAWN assimilation, after two different cases. So perhaps an opening sentence of the paragraph should be something along the lines of: "These findings add to the growing body of evidence that suggests that assimilation of high resolution, high precision Doppler wind lidar profiles into convection allowing models improves analysis of environmental conditions favoring convective storm development and upscale growth. Nevertheless, longer assimilation periods and more flight dates are needed to generalize impacts across a diversity of cases and convective regimes." Then the "this is challenging" and other subsequent statements can remain in place.

Building onto the statements above about CPEX cases, I don't see much motivation in the Intro or text near beginning of the paper for why you selected this particular case, aside from the fact that Turk et al also featured it. Was this one of the few cases where convection developed near in space/time to a region where the DC-8 flew? Some additional description regarding this point would be worthwhile.

Line 568, though you do include the Bedka et al 2020 reference, perhaps a statement about the fact that HALO provides aerosol and water vapor profiles. You make a statement in prior sentences about how aerosol information was lacking in CPEX 2017, and mentioning HALO's capability will close the loop.

Line 567, CPEX-AW is the NASA component of the overall ESA "Tropical Aeolus Cal/Val Campaign", you may want to ping NASA HQ sponsors to determine the best verbiage for describing CPEX-AW. Defining the -AW part in your text would be warranted as well.

General question, how are you assigning observational error to the DAWN wind? Is this based on the results from Greco et al 2020, or are you treating the data like a radiosonde in the assimilation process?

General comment, my impression of ensemble forecasts is that initial conditions are tweaked a tiny bit in each run and the outcome of the ensemble is a probability or likelihood of an event happening based on agreement between the ensemble members. Table 1 indicates this is a 48 member ensemble. But we're only seeing what seems to be one model run from the control and assimilation. Are these ensemble means we're seeing in this paper? I don't see this in the text anywhere but I may have missed something. Some expansion on this in the text would be worthwhile.

General comment 2, do you have any thoughts on why convection was delayed for about an hour in the run with DAWN assimilation. Perhaps you could offer a hypothesis for this in the text? Do you feel that if the DC-8 were on station in this region at say 16 UTC, then the model simulation would have had the correct timing for the convection?

Data Availability, DAWN data is archived at the NASA Langley ASDC: https://asdc.larc.nasa.gov/project/CPEX Is the APR-2 data archived somewhere accessible, rather than the "available upon request" statement?

Acknowledgements, I recommend you credit Michael Kavaya as PI of DAWN and the DAWN team (and Simone Tanelli and the APR-2 team as well) for collecting the data used in your study. "The DC-8 flight support team" does not adequately credit folks associated with the instruments.

---

## Author Comment (AC1) · 8 Feb 2021

**Replies to Anonymous Referee #1**

**"Assimilation of DAWN Doppler Wind Lidar Data During the 2017 Convective Processes Experiment (CPEX): Impact on the Precipitation and Flow Structure"**
**Svetla Hristova-Veleva et al. (AMT-2020-503)**

Comments Line 45, "in the first place" instead of "on the first place"

Line 79, times -> time

*Fixed.*

Line 98, Zhang et al 2018 seems to be related to the NOAA P-3 DWL, not DAWN.

*Actually, this reference is not needed for the point being made, so it has been removed.*

Line 101, recommend defining what you mean by "sparsely-sampled"

*When I wrote that, I was looking at Figure 9, which shows how DAWN profiles are provided more "sparsely" than APR-2 profiles, about a factor of 15. However, the wording is ambiguous and gives the wrong impression. To fix, the paragraph above this has been reworded to reflect this, and I have taken out the "sparsely sampled" description: "The multi-beam measurements from the DAWN lidar were processed into high-resolution vertical wind profiles spaced as finely as 3-7-km along-track (Greco et al., 2020), including the environment close to where the clouds develop".*

Line 135, I'm guessing that the environments weren't 100% cloud free given the tropical environment being sampled. DAWN has the ability to pulse through some tenuous cloud as well. So I'd recommend this being rephrased as "wind profiles in aerosol-rich, clear or broken cloud regions surrounding the convection". You address make a similar statement to what I recommend below in lines 146-147, so it would be good to be consistent with your statements.

*This wording has been replaced as suggested.*

Line 147, recommend "off-nadir" rather than "elevation" which could be interpreted to be 60 degrees coming out of the aircraft by some unfamiliar readers

*The wording is replaced.*

Line 155-158, is GPM data at all assimilated into the boundary condition analyses or coarser grids in your simulation (i.e. is it part of the NCEP "conventional observations" in Table 1, and "clear sky radiances" in line 190)? Or are the simulations entirely independent of GPM, and putting DAWN into the mix made the MCS simulation look a lot more like GPM than the control.

*The simulation (both the control and the one that assimilated data) did not assimilate any GPM observations. Indeed, the assimilation of just the DAWN winds made the MCS simulation become much closer to the GPM observations. Please, see the new text first paragraph of Section 3) which now includes: "**The GPM data were not assimilated in this study. Neither were the APR-2 radar data. Both of these datasets were only used for model validation. The only assimilated data, in addition to the NCEP conventional observations, were the DAWNv3 wind profiles. Hence, the improved representation of the precipitation structure in the simulations with data assimilation is solely the results of assimilating the DAWN winds**."*

Line 188, Blue seems like it is depicting both areas with little aerosol and cloud obscuration. You should consider depicting the two sources of DAWN data dropout with differing colors to better inform the readers.

*The intent of that figure is only to show the cross section of the DAWN data available to the data assimilation system during the one-hour period. The reasons for the dark colored area are could be as you state above, but distracts from the point of the figure. Figures 8, 12, 15 and 18 of the earlier Turk et al 2020 paper show the locations of each DAWN vector against the APR-2 Ka-band radar profile, which gives the reader a good sense of where the clouds are in relation to the "missing" DAWN data, the low-aerosol conditions.*

Figure 2, vertical axis should be made the same across all 4 panels to be consistent and to enable comparisons across segments

*The figure has been changed as suggested.*

Page 13-14, I don't feel that you've adequately explained the nuances associated with Ka and Ku band reflectivity. Some places Ku only is shown (Fig 5), others both bands are shown. It would be helpful to provide a couple sentence explanation of the characteristics of these multiple wavelength data.

*We have added in some wording to the differences between precipitation estimated by a single-vs dual-frequency precipitation radar, but the Grecu et al reference explains this in detail.*

Line 316, is this 1.8 km "blind spot" common for APR-2 and -3 instruments, or was this due to a instrument specific scan mode setting at the time of this flight?

*This is due to the pulse repetition frequency (PRF) setting for the APR-2 at that time.*

Line 328, delete the 11a that has been crossed out.

*Fixed.*

Figures 7-12, though it is demonstrated that the control generated almost no precip where there should have been some, I recommend you explicitly state somewhere that the profiles are from the run with DAWN assimilation.

*That is a good point to highlight, as it may not be obvious to many readers.   We have added wording to this point right at the beginning of Section 4.*

Figure 16, I'm not seeing observational evidence of enhanced moisture, seems to look more like precipitation features rather than a moisture image. I recommend you reconsider this figure and the text that goes with it.

*This was a mistake on our part.  We appreciate this oversight being caught.  When that figure was made into one panel, the wording should have been updated to remove the mention of the total precipitable water (also available at that same URL).We have updated the figure to show both the rain index and the total precipitable water (TPW) side by side.*

Figure 18, should the temperature anomaly in color shading in Figure 18b be aligned in some sense with the cold pool boundary? I am seeing a disconnect and am confused. I realize that the dropsondes revealed a cold pool there but I'm guessing a more substantial cold pool would have been found near the precip in the lower right quadrant of the domain had you released dropsondes there. Perhaps the red line is not particularly important because it confuses the interpretation of Fig 18b,

*The red line is the boundary that was taken from the Zipser and Rajagopal analysis of this same flight date (shown in Figure A below), using their analysis of dropsonde and GPM IMERG data. We feel that this boundary does give overall context to the "after-assimilation" depiction, as the cold pool signature is a manifestation of temperature and moisture conditions.  Granted, this is based on limited observational data Since the analysis is limited to only a few dropsondes on this particular day, it's entirely possible that a more substantial cold pool would have been revealed further south.   However, in contrast to the simulations, the observations did not reveal a precipitating system further south.*

[Figure]

**Figure A.** "Hand-drawn" dropsonde analysis from Zipser and Rajagopal (2018).

Line 547, the sentence beginning with "It resulted in modification of near surface convergence" seems a lot like line 549, "it led to increase of the near surface convergence". Was this an inadvertent repeat, or are there important nuances in the transition between the two sentences that I'm not understanding?

*The order of those two sentences is reversed now, making more sense: "Analysis of individual variables revealed that the assimilation of the DAWN winds resulted in important and coherent modifications of the environment. It resulted in modification of the near surface convergence (increased), the 2-m air temperature (increased) and the water vapor (increased)."*

Line 550, you state that convection was not present in the control run, and then line 553 you state that precipitation was not present in the control run. Along the lines of the previous comment, perhaps you could review the paragraph beginning with line 545 and streamline the messaging. It's a very important paragraph to your summary so it's necessary to be clear with your statements.

*We have removed the wording "not present in the control run"- the sentence now reads, "It led to increase of the near surface convergence, temperature and water vapor, creating more*

*favorable conditions for the development of convection exactly where it was observed"*, which is the point we want to make.

Line 560, you note that this was a single case, but the final sentence of the previous paragraph, you note that Cui et al found similar results as you, in that convection was better predicted through DAWN assimilation, after two different cases. So perhaps an opening sentence of the paragraph should be something along the lines of: "These findings add to the growing body of evidence that suggests that assimilation of high resolution, high precision Doppler wind lidar profiles into convection allowing models improves analysis of environmental conditions favoring convective storm development and upscale growth. Nevertheless, longer assimilation periods and more flight dates are needed to generalize impacts across a diversity of cases and convective regimes." Then the "this is challenging" and other subsequent statements can remain in place.

*That's a well written statement and we have inserted it (slight tense changes) right at the beginning of the last paragraph in the conclusions section. However, we do want to stress that the impact is felt not just in the wind per-se, but in particular the representation of the wind* **derivative** *field, that is the divergence (or convergence, the opposite sign of divergence). Both the Cui et al study and this one noticed this unique finding. This is the important connection to vertical growth.*

Building onto the statements above about CPEX cases, I don't see much motivation in the Intro or text near beginning of the paper for why you selected this particular case, aside from the fact that Turk et al also featured it. Was this one of the few cases where convection developed near in space/time to a region where the DC-8 flew? Some additional description regarding this point would be worthwhile.

*This was one of the dates from CPEX where the precipitation appeared to develop from localized forcing, less influenced by the large scale forcing (the precipitation that was well north of the AOI). Of course, in nature the two are superimposed, but this case was somewhat isolated with its vertical structure of convection, the DC-8 could maneuver around it, and happened to be on-station. Also, the CPEX PI's had set up a flight pattern for that day boxed in (not ideally of course, as the pilots have authority to maneuver as needed for safety reasons) the cloud development. The materials shown in the T2020 paper were intentionally designed so as to be pointed back to within this second follow-on paper.*

Line 568, though you do include the Bedka et al 2020 reference, perhaps a statement about the fact that HALO provides aerosol and water vapor profiles. You make a statement in prior sentences about how aerosol information was lacking in CPEX 2017, and mentioning HALO's capability will close the loop.

*Good suggestion and this sentence is added, "The HALO instrument provides aerosol and water vapor profiles, to complement the DAWN capability."*

Line 567, CPEX-AW is the NASA component of the overall ESA "Tropical Aeolus Cal/Val Campaign", you may want to ping NASA HQ sponsors to determine the best verbiage for describing CPEX-AW. Defining the -AW part in your text would be warranted as well.

*We have dialed back the CPEX-AW 2021 date, and removed emphasis on the "synchronous" observations with ADM-Aeolus just in case those satellite data are unavailable. The statement now reads, "The proposed NASA-ESA CPEX-AW field campaign will provide the opportunity to fly the APR-3, DAWN and the HALO lidar (Bedka et al., 2020) alongside available ADM-Aeolus observations, in the eastern Atlantic where African easterly waves interact with the Saharan air layer (Zipser et al., 2009). The HALO instrument provides aerosol and water vapor profiles (a missing component of these airborne data), to complement the DAWN wind-sensing capability."*

General question, how are you assigning observational error to the DAWN wind? Is this based on the results from Greco et al 2020, or are you treating the data like a radiosonde in the assimilation process?

*Taking reference from radiosonde data errors, the observation error standard deviation for DAWN wind is prescribed as 1 (m/s) at the sampling time centered at analysis time. Observation error standard deviation increases linearly corresponding to the sampling time away from analysis time. A quality control procedure rejects data when the observation departure is bigger than 8 (m/s).*

General comment, my impression of ensemble forecasts is that initial conditions are tweaked a tiny bit in each run and the outcome of the ensemble is a probability or likelihood of an event happening based on agreement between the ensemble members. Table 1 indicates this is a 48-member ensemble. But we're only seeing what seems to be one model run from the control and assimilation. Are these ensemble means we're seeing in this paper? I don't see this in the text anywhere but I may have missed something. Some expansion on this in the text would be worthwhile.

*The NU-WRF EDAS consists of ensemble forecasts and a central forecast. The ensemble forecasts are used to estimate the flow-dependent background error covariance. The analysis updates the initial conditions for the subsequent central forecast that is not the ensemble mean, though very close to it. The analysis error covariance generates the ensemble perturbations for the ensemble forecasts in the next cycle. The results presented in the paper are from the central analyses and forecasts.*

General comment 2, do you have any thoughts on why convection was delayed for about an hour in the run with DAWN assimilation. Perhaps you could offer a hypothesis for this in the text? Do you feel that if the DC-8 were on station in this region at say 16 UTC, then the model simulation would have had the correct timing for the convection?

*There could be a number of reasons for this. Given that the impact of upon the precipitation structure resulted from sampling earlier in time, prior to onset, your suggestion is certainly not unreasonable. The DC-8 approached the AOI from the west in large part because that was the the direction the DC-8 was heading upon transit from Florida. It's certainly plausible; sampling "upstream" of where precipitation develops has been highlighted for its impact upon mesoscale weather systems, e.g., see the review by Majumdar et al.:*

S. J. Majumdar, "A Review of Targeted Observations," *Bulletin of the American Meteorological Society*, vol. 97, no. 12, pp. 2287–2303, Dec. 2016, doi: 10.1175/BAMS-D-14-00259.1.

*Another possible explanation is that the 1-hourly update cycle is not "fine enough" in time to resolve such a fast evolution.   From this single case, however, there is insufficient evidence to point to one or the other of these explanations.*

Data Availability, DAWN data is archived at the NASA Langley ASDC: https://asdc.larc.nasa.gov/project/CPEX Is the APR-2 data archived somewhere accessible, rather than the "available upon request" statement?

*For CPEX, we added the URL for the APR2 archive and also this one for the DAWN data.*

Acknowledgements, I recommend you credit Michael Kavaya as PI of DAWN and the DAWN team (and Simone Tanelli and the APR-2 team as well) for collecting the data used in your study. "The DC-8 flight support team" does not adequately credit folks associated with the instruments.

*We have added these statements.  "The DC-8 flight support team at NASA Armstrong Flight Research Center" is clarified- they handled all-around DC-8 flight operations including arrangement of instrument integration.*

---

## Referee Comment (RC2) · Anonymous Referee #2 · 23 Feb 2021

General comments:

This manuscript investigates the impact of the assimilation of DAWN derived winds in the NASA NU-WRF EDAS data assimilation system on the precipitation forecast structure. Benefits are analyzed by investigating direct forecasts of the precipitation field as well as by performing a detailed study of the analysis increments in terms of moisture, temperature and wind fields. Results are verified against APR-2 and GPM satellite observations available during the time of the experiment.

[Figure]

Although results are limited to the analysis of a single flight event (one day of data assimilation experiments), this manuscript is well organized, and results are discussed with a great level of detail. I recommend the publication of this study after a couple of clarifying questions listed below are addressed.

Specific comments:

Coming from a data assimilation background, I was sometimes confused by the use of "simulation" and "assimilation". Is the word "simulation" used for verification purposes only rather than for the use of a forward model to perform data assimilation?

Also, it is not clear whether the assimilation is conducted in the regional system, in the global model that provides initial and boundary conditions, or in both. Was the assimilation just conducted in the NCEP's data assimilation and forecast system? If the answer is "yes", did you attempt to conduct direct data assimilation into the NU-WRF system? If the answer is "no", in which of the two domains was the assimilation conducted? Was it a 2-way nested configuration? If would be interesting to investigate the impact of the assimilation of the observations in the WRF system as well as to analyze the benefits that come directly from the assimilation in the global system through improved initial and boundary conditions.
* * *

---

## Author Comment (AC2) · 23 Feb 2021

**Replies to Anonymous Referee #2**

**"Assimilation of DAWN Doppler Wind Lidar Data During the 2017 Convective Processes Experiment (CPEX): Impact on the Precipitation and Flow Structure"**
**Svetla Hristova-Veleva et al.  (AMT-2020-503)**

**General comments:**

This manuscript investigates the impact of the assimilation of DAWN derived winds in the NASA NU-WRF EDAS data assimilation system on the precipitation forecast structure. Benefits are analyzed by investigating direct forecasts of the precipitation field as well as by performing a detailed study of the analysis increments in terms of moisture, temperature and wind fields. Results are verified against APR-2 and GPM satellite observations available during the time of the experiment.

Although results are limited to the analysis of a single flight event (one day of data assimilation experiments), this manuscript is well organized, and results are discussed with a great level of detail. I recommend the publication of this study after a couple of clarifying questions listed below are addressed.

**Specific comments:**

Coming from a data assimilation background, I was sometimes confused by the use of "simulation" and "assimilation". Is the word "simulation" used for verification purposes only rather than for the use of a forward model to perform data assimilation?

*In this paper, a simulation refers to the WRF model forward integration, which may/may not have the initial conditions corrected by DAWN observations assimilated.  Whereas assimilation is the procedure that combines the information in DAWN observations and in so-called "background" states from a forward model forecast to produce a corrected initial condition for further model forward integration.*

Also, it is not clear whether the assimilation is conducted in the regional system, in the global model that provides initial and boundary conditions, or in both. Was the assimilation just conducted in the NCEP's data assimilation and forecast system? If the answer is "yes", did you attempt to conduct direct data assimilation into the NU-WRF system? If the answer is "no", in which of the two domains was the assimilation conducted? Was it a 2-way nested configuration? If would be interesting to investigate the impact of the assimilation of the observations in the WRF system as well as to analyze the benefits that come directly from the assimilation in the global system through improved initial and boundary conditions.

*The lateral boundary conditions from NCEP already have the conventional data and all other operational data streams assimilated (DAWN data is **not** part of this).  This is a standard and*

*necessary procedure for a regional model to run. In this paper, the assimilation is conducted in the regional system: NU-WRF EDAS (developed at NASA Goddard, see the brief description in the manuscript, and references within). The ensemble data assimilation (using conventional observations **and** DAWN observations) is carried out in the domain 1. The WRF model forward integration is configured as 1-way nesting. When the regional model integrates forward, the domain **interior** states evolve differently and could drift away comparing to the global analysis. Thinking this way, the data impact (such as from the conventional data) at the boundary is lost in the domain interior, thus justifying the existence of regional data assimilation in the domain (i.e., no assimilation at or near the boundary). Technically, one could say that the conventional data are thereby "assimilated twice". A more meaningful way would be to view this as "a re-enforcement of the data constraint in the regional model interior". Regarding the last point, since DAWN data is from a short field campaign period and very limited spatial coverage, it is beyond the scope of this work to investigate the data impact in a global assimilation system.*